# Comparison of nebivolol versus diltiazem in improving coronary artery spasm and quality of life in patients with hypertension and vasospastic angina: A prospective, randomized, double-blind pilot study

Hyungdon Kook[1], Soon Jun Hong[1]*, Kyung-Sook Yang[2], Sunki Lee[3], Jung-Sun Kim[4], Chang Gyu Park[5]*

1 Division of Cardiology, Department of Internal Medicine, Korea University Anam Hospital, Korea University College of Medicine, Seoul, Korea, 2 Department of Biostatistics, Korea University College of Medicine, Seoul, Korea, 3 Division of Cardiology, Department of Internal Medicine, Dongtan Sacred Heart Hospital, Hallym University College of Medicine, Hwaseong, Korea, 4 Division of Cardiology, Department of Internal Medicine, Severance Cardiovascular Hospital, Yonsei University College of Medicine, Seoul, Korea, 5 Division of Cardiology, Department of Internal Medicine, Korea University Guro Hospital, Korea University College of Medicine, Seoul, Korea

* psyche94@gmail.com (SJH); parkcg@kumc.or.kr (CGP)

## Abstract

### Background

Beta-blockers are often not the preferred treatment for patients with vasospastic angina. However, nebivolol, beta-blocker with nitric oxide-releasing effect, could theoretically improve coronary vasospasm. We compared nebivolol versus diltiazem in improving coronary vasospasm and quality of life in patients with hypertensive vasospastic angina during a 12-week follow-up.

### Methods

Fifty-one hypertensive patients with documented coronary vasospasm were randomly allocated into 3 treatment groups: (1) Nebivolol Group (5mg for 2 weeks/10mg for 10 weeks); (2) Diltiazem Group (90mg for 2 weeks/180mg for 10 weeks); (3) Low-dose Combination Group (2.5mg + 45mg for 2 weeks/5mg + 90mg for 10 weeks). The primary endpoint was to compare the percent changes in coronary vasospasm at 12 weeks from baseline among the 3 groups. The secondary endpoints included changes in quality of life based on the Seattle Angina Questionnaire and changes in blood pressure at 12 weeks from baseline.

### Results

Significant improvements in coronary vasospasm were found in all groups; however, the improvement in percent changes in coronary artery spasm was greatest in the Diltiazem Group (50.4±8.8% vs. 67.8±12.8% vs. 46.8±12.3%, Nebivolol Group vs. Diltiazem Group p = 0.008; Nebivolol Group vs. Low-dose Combination Group p = 0.999; Diltiazem Group vs.

**Data Availability Statement:** All relevant data are within the paper and its Supporting Information files.

**Funding:** SJH received funding from Bio & Medical Technology Development Program of the National Research Foundation (NRF) funded by the Ministry of Science and ICT [NRF-2018M3A9A8017949]. The funder had no role in study design, data collection and analysis, decision to publish, or preparation of the manuscript.

**Competing interests:** The authors have declared that no competing interests exist.

Low-dose Combination Group p = 0.017). The overall Seattle Angina Questionnaire scores were significantly elevated at 12 weeks compared to the baseline in entire study population. There were no significant differences between the three groups in the overall Seattle Angina Questionnaire score changes and blood pressure changes.

## Conclusions

Both nebivolol and diltiazem showed significant coronary vasospasm reduction effect, but the effect was greater for diltiazem.

## Introduction

Vasospastic angina occurs at rest and is caused by a coronary artery spasm with dynamic electrocardiogram changes and preserved exercise capacity [1]. Though the long-term prognosis of patients with vasospastic angina is known to be good [2], vasospastic angina can reduce the quality of life by causing severe angina, and more seriously result in myocardial infarction, lethal arrhythmia, and sudden cardiac death in rare cases [3]. Calcium channel blockers are an established first-line pharmacotherapy for vasospastic angina because of their relaxation effects on coronary artery smooth muscles [4, 5]. In contrast, the usage of early generation beta-blockers is considered to be a contraindication in patients with vasospastic angina, as their use could aggravate a coronary artery spasm by leaving alpha-mediated vasoconstriction despite beta-mediated vasodilation [4].

The correlation between endothelial dysfunction and the risk of coronary heart disease is well known through previous studies [6, 7]. The reduction of nitric oxide synthesis, nitric oxide degradation due to oxidative stress, and decreased nitric oxide sensitivity to vascular dilating effects contribute to the pathogenesis of vasospastic angina [8, 9]. In particular, patients with high blood pressure have been known to have impaired vascular endothelial function, mainly due to increased biomechanical friction in the arterial endothelium and decreased biological availability of nitric oxide [10]. A recent report showed that severe endothelial dysfunction may lead to the progression of coronary artery disease and future adverse cardiovascular events [11]. However, the medical treatments available for patients with hypertensive vasospastic angina who have endothelial dysfunction are quite limited. As mentioned, beta-blockers have been reported to inhibit vascular dilatation of adrenaline stimuli, such as propranolol exacerbating vasospastic angina [12]. However, a series of reports on the vascular dilatation of the latest third-generation beta-blockers provide new information on the role of beta-blockers in patients with vasospastic angina. Furthermore, nebivolol is known to selectively block the endothelial β-1 adrenaline receptor to create vascular dilatation, and also to stimulate β-3 adrenaline receptors to increase nitric oxide generation and provide antioxidant effects in the vascular endothelium [13, 14]. Furthermore, patients who have been treated with nebivolol may benefit from a better quality of life [15]. Therefore, we performed a prospective, randomized, double-blind pilot study to compare nebivolol versus diltiazem in improving coronary artery spasm and quality of life in hypertensive patients with vasospastic angina during a 12-week follow-up.

## Methods

### Study population

Patients 35–80 years of age were eligible based on the following criteria: (1) diagnosed with stage 1–2 hypertension (systolic blood pressure, 140–179 mmHg; and diastolic blood pressure,

90–109 mmHg); and (2) had resting angina and diagnosed with vasospastic angina through coronary angiography and provocation test (insignificant coronary artery stenosis with positive provocation test). A total of 260 patients were screened for inclusion at Korea University Anam Hospital, Guro Hospital, Ansan Hospital, and Severance Hospital from January 2018 to March 2019 (Fig 1). The exclusion criteria were as follows: (1) previous history of hypersensitivity to beta-blockers or calcium channel blockers; (2) history of dementia or accompanying psychiatric illness or history of drug abuse; (3) those who participated in other clinical trials less than 1 month before the screening; (4) study subjects who were taking drugs that can affect the evaluation of the study drug efficacy (beta-blockers, calcium channel antagonists, nicorandil, and nitrates), but these subjects were allowed to participate after a wash-out period of at least 2 weeks; (5) malignant hypertension (with retinal hemorrhage or papilledema) or known moderate or severe retinopathy (retinal hemorrhage within the last 6 months, visual disturbance, or retinal microaneurysm); (6) a history of secondary hypertension and all suspected secondary hypertension including coarctation of the aorta, hyperaldosteronism, renal artery stenosis, Cushing's disease, chromatin-positive cell tumor, or polycystic kidney disease; (7) orthostatic hypotension with symptoms; (8) heart disease (left ventricular ejection fraction $<$ 50%), ischemic heart disease (any significant coronary artery stenosis ($>$50%) in coronary angiography, angina pectoris or myocardial infarction), percutaneous coronary intervention, or coronary artery bypass surgery; (9) severe cerebrovascular disease (stroke, cerebral infarction, or cerebral hemorrhage within the last 6 months); (10) serum creatinine $>$ 1.5 mg/dL; (11) severe liver failure, aspartate aminotransferase or alanine aminotransferase $>$ 3 times the normal upper limit, biliary obstruction, biliary cirrhosis, or cholestasis; (12) gastrointestinal diseases, patients undergoing surgery that may affect the absorption, distribution, metabolism, and excretion of drugs, current active gastritis and gastrointestinal/rectal bleeding that the tester considered clinically significant, or active inflammatory bowel syndrome within the last 12 months; (13) pregnant and lactating women, or those who had a pregnancy plan during the trial and did not agree with an appropriate method of contraception.

## Study design

The present study was a prospective, randomized, double-blind trial. A total of 51 hypertensive patients with documented coronary vasospasm were equally and randomly allocated into 3 treatment groups: (1) oral nebivolol (5 mg) once a day for two weeks, followed by nebivolol (10 mg) once a day for 10 weeks (Nebivolol Group); (2) oral slow-releasing diltiazem (90 mg) once a day for 2 weeks, followed by slow-releasing diltiazem (180 mg) once a day for 10 weeks (Diltiazem Group); and (3) oral nebivolol (2.5 mg) and slow-releasing diltiazem (45 mg) once a day for 2 weeks, followed by nebivolol (5 mg) and slow-releasing diltiazem (90 mg) once a day for 10 weeks (Low-dose Combination Group) (Fig 1).

The Nebivolol group and the Diltiazem group had a placebo drug to balance the total number of tablets in all 3 groups. The Nebivolol group was additionally given a placebo with the same color, shape, and size as diltiazem, and the Diltiazem group was additionally given a placebo with the same color, shape, and size as nebivolol. The Low-dose combination group was given by preparing a half-dose drug in the same size as the original dose. These study drugs were packaged in identical containers. The attending physician, study subject, personnel at investigative sites, and all study staff and investigators, other than the safety monitoring board were blinded to treatment assignment via a locked web-based system. Patients who had complained of chest pain with a left anterior descending coronary artery spasm $>$ 75% after the acetylcholine provocation test and had agreed to participate in the study were centrally randomly assigned to a computer-generated random assignment number in a 1:1:1 ratio,

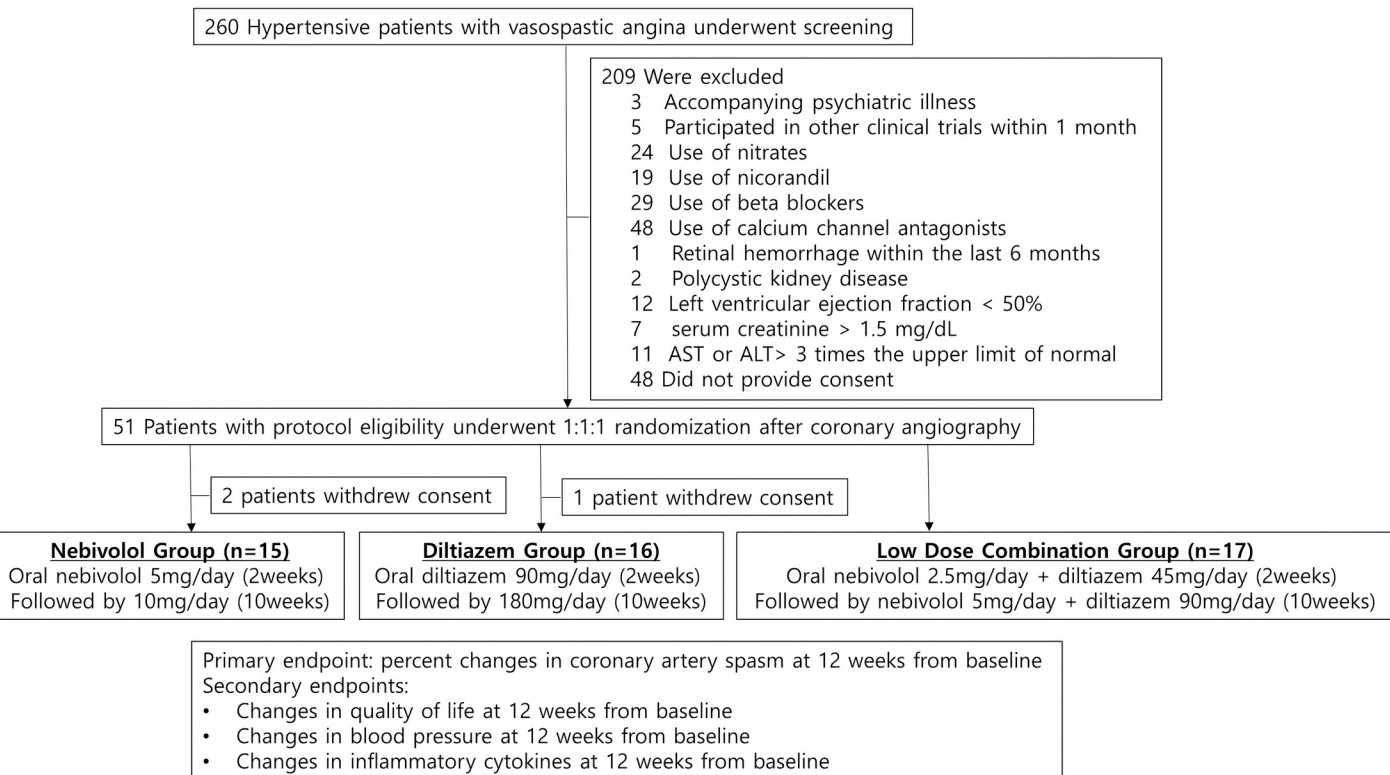

**Fig 1. Study design.** Patients were randomly categorized into 3 treatment groups: (1) oral nebivolol 5mg once a day for two weeks, followed by nebivolol 10mg once a day for 10 weeks (Nebivolol Group); (2) oral diltiazem 90mg once a day for 2 weeks, followed by diltiazem 180mg once a day for 10 weeks (Diltiazem Group); (3) oral nebivolol 2.5mg and diltiazem 45mg once a day for 2 weeks followed by nebivolol 5mg and diltiazem 90mg once a day for 10 weeks (Low-dose Combination Group) AST = aspartate aminotransferase; ALT = alanine aminotransferase.

according to the randomization table. Randomization was balanced with a block randomization method. The web-based computer-generated randomization table was developed by an external programmer who was not otherwise involved in the trial. This study was approved by the Korea University Hospital Institute Review Board (IRB No. 2015AN0296) from January 1, 2018 to December 31, 2019, and written informed consent was obtained from all participants or their legal representatives. Complete date range for patient recruitment was from January 2018 to March 2019. Follow-up duration for this study was 12 weeks. The study also complied with the Declaration of Helsinki. This study was registered in ClinicalTrials.gov (Name of the registry: The Effect of Nebivolol in Hypertensive Patients With Coronary Arterial Spasm; Trial number: NCT03930433; Trial URL: https://clinicaltrials.gov/ct2/show/NCT03930433?term= NCT03930433&draw=2&rank=1).

There was a delay in registering this study after enrolment of participant started due to funding issues which have nothing to do with the protocol itself. The authors confirm that all ongoing and related trials for these drugs are registered.

## Endpoints

The primary endpoint was to compare percent changes in coronary artery spasm after intracoronary acetylcholine administration at 12 weeks from baseline among the 3 groups. The secondary endpoints included changes in quality of life based on the Seattle Angina Questionnaire and changes in blood pressure at 12 weeks from baseline [16]. The Seattle Angina

Questionnaire was obtained at baseline and at 12 weeks. Blood pressure, including systolic, diastolic, and pulse pressure, was measured at baseline, 2 weeks, 6 weeks, and 12 weeks.

## Coronary angiography and vasospasm provocation test

Coronary angiography was performed at baseline and repeated 12 weeks after randomization. Coronary angiography was performed by engaging a Judkins catheter after puncturing the radial artery or femoral artery. Before the coronary angiography, an electrocardiogram was attached to record the changes in the electrocardiogram during the acetylcholine provocation test. The dose of acetylcholine was gradually increased in the order of 20 µg, 50 µg, and 100 µg in the left coronary artery, gradually injected over 60 seconds, and each dose of acetylcholine was administered every 3 minutes [17]. If the induction test was terminated or if acetylcholine had provoked significant coronary artery vasospasm, chest pain, or electrocardiographic changes, 200 µg of nitroglycerin was administrated into the coronary artery. The percent changes in the coronary diameter were measured with a computerized quantitative analyzer by measuring the internal diameter of the artery using a caliper. The positive coronary artery spasm was defined as a transient luminal narrowing > 75% and ischemic electrocardiography findings (elevation of the ST segment or a depression of more than 1 mm, inversion of T wave) with chest pain.

## Seattle Angina Questionnaire

The Seattle Angina Questionnaire was used for evaluating the quality of life associated with angina. The questionnaire consisted of 19 items in 5 categories. The score was calculated from 0 (worst) to 100 (best) for each category. In this study, the questionnaire was conducted before the start of medication and at the end of the 12-week random treatment.

## Laboratory analysis of inflammatory markers and lipid profiles

Venous blood samples were drawn from patients after fasting for eight hours or overnight before performing index coronary angiography. Follow-up samples were drawn after at the end of the 12-week random treatment before follow-up coronary angiography. Blood samples were centrifuged to obtain plasma that was stored at −80˚C. Inflammatory markers, such as high-sensitivity C-reactive protein (hsCRP), interleukin-6 (IL-6), tumor necrosis factor- α (TNF-α), adiponectin, soluble intercellular adhesion molecule-1 (sICAM-1), and soluble vascular cell adhesion molecule-1 (sVCAM-1), were measured at the beginning of the study and at the 12-week follow-up. TNF-α was measured by a sandwich enzyme-linked immunosorbent assay (ELISA) with a minimum detectable level of 0.5 pg/mL (ALPCO Diagnostics, Salem, NH, USA). IL-6 was also measured by a sandwich ELISA with a minimum detectable level of 0.16 pg/mL (ALPCO Diagnostics, Salem, NH, USA). The hsCRP concentrations were quantified using a latex nephelometer II (Dade Behring Inc., Newark, DE, USA). The plasma adiponectin concentration was assessed using a radioimmunoassay (Linco Research Inc., St. Charles, MO, USA). The sensitivity of this assay was 0.78 ng/mL. The coefficients of variation for the intra- and inter-assay were 9.3% and 15.3%, respectively. Furthermore, sICAM-1 and sVCAM-1 were measured using a commercially available ELISA according to the manufacturer's instructions (R&D Systems, Minneapolis, MN, USA). Total cholesterol, triglyceride, high-density lipoprotein cholesterol, and low-density lipoprotein cholesterol levels were determined using enzymatic methods with standard biochemical procedures on a BM Hitachi automated clinical chemistry analyzer (Hitachi, Tokyo, Japan).

## Statistical analysis

Continuous variables were expressed as mean ± standard deviation. Categorical variables were expressed as the number and percentage of patients. The baseline demographics comparisons between three groups were performed by using analysis of variance (ANOVA) for normal distribution or Kruskal-Wallis test for non-normal distribution. When the result of ANOVA or Kruskal-Wallis test was significant (p-value<0.05), post-hoc analysis was performed using Bonferroni's correction, Tukey-Kramer test, and Dunnett's T3 test as appropriate. The linear mixed model was used for comparing baseline levels of primary endpoint and secondary endpoints between groups, estimating change over time within groups, and determining if change over time is significantly different between groups. The linear mixed model included fixed effects for the three-level group indicator, the two-level time indicator, and an interaction effect, and some covariates. The covariates included for fitting the model were study group, age, sex, primary and secondary endpoints, and baseline demographic variables that was less than p<0.1 when comparing between groups. Changes over time were calculated as the difference in the values at the end of the 12-week treatment and at baseline. Per-protocol analysis was performed in this study. All tests were two-tailed with a P-value less than 0.05 considered as significant. To the best of our knowledge, this was the first pilot study to compare nebivolol versus diltiazem in improving coronary artery spasm; therefore, the sample size could not be calculated from previously published data. All statistical analyses were carried out using SPSS ver. 24.0 (SPSS-PC Inc., Chicago, Illinois, USA).

## Results

### Baseline characteristics of patients

The baseline characteristics of the study population (n = 48) are shown in Table 1. The baseline characteristics of mean age, sex distribution, body mass index, low-density lipoprotein-cholesterol, high-density lipoprotein-cholesterol, and triglyceride between the three groups were similar. Total cholesterol was higher in the Low-dose Combination Group than the Diltiazem Group (Nebivolol Group vs. Diltiazem Group vs. Low-dose Combination Group: 187.7±49.1 mg/dL vs. 178.4±27.3 mg/dL vs. 212.1±39.0 mg/dL, p = 0.040). The frequency of smoking and diabetes in the Nebivolol group was numerically high but not statistically significant. The frequency of social drinking tended to be lower in the Combination group than it was in the Nebivolol group (Odds ratio 0.222, 95% CI 0.046–1.083, p = 0.076).

### Coronary angiographic findings at baseline and at the 12-week follow-up

In baseline coronary angiography, significant vasospasm after intracoronary acetylcholine administration in linear mixed model was observed in all three groups without statistical differences (percent changes in diameter in the Nebivolol Group vs. Diltiazem Group vs. Low-dose Combination Group:-87.9±4.9% vs. -86.4±6.3% vs. -85.5±5.7%, Nebivolol Group vs. Diltiazem Group p = 0.977; Nebivolol Group vs. Low-dose Combination Group p = 0.838; Diltiazem Group vs. Low-dose Combination Group p = 0.997) (Table 2). All patients had no fixed coronary artery lesion stenosis >50% in coronary angiography. In the follow-up coronary angiography after 12 weeks of study medications, a significant improvement in coronary vasospasm was observed in all three groups (Nebivolol Group -37.5±6.4%, Diltiazem Group -18.8 ±9.4%, Low-dose Combination Group -38.9±8.6%, all p<0.001), but the magnitude of improvement in vasospasm was the greatest in the Diltiazem Group (percent changes in diameter from the baseline provocation test for Nebivolol Group vs. Diltiazem Group vs. Low-dose Combination Group: 50.4±8.8% vs. 67.8±12.8% vs. 46.8±12.3%, Nebivolol Group vs. Diltiazem

**Table 1. Baseline characteristics of the study groups.**

| | Nebivolol group (n = 15) | Diltiazem group (n = 16) | Combination group (n = 17) | p-value |
|---|---|---|---|---|
| Age | 63.4±7.4 | 59.5±11.8 | 62.8±7.2 | 0.538 |
| Male sex | 12 (80.0) | 11 (68.8) | 9 (52.9) | 0.107 |
| Smoking history | 6 (40.0) | 4 (25.0) | 4 (23.5) | 0.319 |
| Social drink | 12 (80.0) | 11 (68.8) | 8 (47.1) | 0.053 |
| BMI | 25.2±3.1 | 24.3±2.8 | 24.0±2.3 | 0.461 |
| Diabetes Mellitus | 3 (20.0) | 1 (6.3) | 2 (11.8) | 0.508 |
| Dyslipidemia | 10 (66.7) | 10 (62.5) | 9 (52.9) | 0.429 |
| Hemoglobin, g/dL | 14.5±1.2 | 14.9±1.7 | 14.5±1.4 | 0.615 |
| Platelet count, ×$10^3$/μL | 209.7±31.5 | 235.6±68.2 | 219.4±52.4 | 0.400 |
| WBC, ×$10^3$/μL | 6.59±1.43 | 7.03±1.13 | 6.30±0.81 | 0.186 |
| K, mmol/L | 4.4±0.4 | 4.4±0.5 | 4.3±0.2 | 0.930 |
| Creatinine, mg/dL | 0.93±0.22 | 0.86±0.13 | 0.78±0.13 | 0.066 |
| AST, IU/L | 29.1±12.1 | 29.2±15.1 | 22.9±2.4 | 0.083 |
| ALT, IU/L | 26.9±11.8 | 26.1±17.9 | 21.1±4.7 | 0.231 |
| ALP, IU/L | 75.5±18.8 | 71.7±27.4 | 73.1±16.8 | 0.894 |
| Bilirubin, total, mg/dL | 0.82±0.42 | 0.82±0.32 | 0.82±0.32 | 0.998 |
| Albumin, g/dL | 4.6±0.7 | 4.6±0.6 | 4.4±0.2 | 0.568 |
| Cholesterol, total, mg/dL | 187.7±49.1 | 178.4±27.3 | 212.1±39.0 | 0.040 |
| Triglyceride, mg/dL | 131.6±102.3 | 160.1±88.4 | 152.5±67.6 | 0.662 |
| HDL, mg/dL | 56.1±16.1 | 49.1±15.3 | 52.7±15.9 | 0.494 |
| LDL, mg/dL | 114.9±41.8 | 111.9±24.3 | 140.2±35.1 | 0.065 |
| Glucose, mg/dL | 104.9±22.7 | 120.4±47.4 | 99.5±11.2 | 0.239 |

Values are presented as number of patients (%) or mean±standard deviation. BMI, body mass index; WBC, white blood cell; BUN, blood urea nitrogen; AST, aspartate aminotransferase; ALT, alanine aminotransferase; ALP, alkaline phosphatase; HDL, high-density lipoprotein; LDL, low-density lipoprotein.

Group p = 0.008; Nebivolol Group vs. Low-dose Combination Group p = 0.999; Diltiazem Group vs. Low-dose Combination Group p = 0.017) (Table 3, Fig 2). None of the patients who completed follow-up coronary angiography at 12 weeks after randomization had more than 75% of vasospasm.

## Changes in quality of life during the 12-week follow-up

The changes in the Seattle Angina Questionnaire during the 12-week follow-up is shown in Table 4. Details regarding components of Seattle Angina Questionnaire are shown in S1 Table. After 12 weeks of study medications, a significant improvement in quality of life based on the

**Table 2. Coronary angiographic changes during the 12-week follow-up.**

| | Baseline CAG | | | | | 12-week follow-up CAG | | | | p-value[a] |
|---|---|---|---|---|---|---|---|---|---|---|
| | Vessel diameter, mm | | Vasospasm | | | Vessel diameter, mm | | Vasospasm | | |
| | IC NTG | IC ACH | mm | % | | IC NTG | IC ACH | mm | % | |
| Nebivolol (n = 15) | 2.98±0.54 | 0.37±0.17 | -2.61±0.44 | -87.9±4.9 | (n = 14) | 3.06±0.42 | 1.90±0.26 | -1.16±0.28 | -37.5±6.4 | <0.001 |
| Diltiazem (n = 16) | 3.07±0.49 | 0.42±0.22 | -2.65±0.44 | -86.4±6.3 | (n = 13) | 3.14±0.38 | 2.56±0.49 | -0.58±0.28 | -18.8±9.4 | <0.001 |
| Combination (n = 17) | 2.93±0.68 | 0.43±0.20 | -2.50±0.60 | -85.5±5.7 | (n = 15) | 2.87±0.57 | 1.75±0.42 | -1.13±0.37 | -38.9±8.6 | <0.001 |

Values are presented as number of patients (%) or mean±standard deviation. CAG, coronary angiography; IC, intra-coronary; NTG, nitroglycerin; ACH, acetylcholine.
[a] p-value was obtained by Tukey-Kramer test for the interaction effect between time and group for diameter percent change in linear mixed model.

**Table 3. Vessel diameter changes between baseline and 12-week follow-up.**

|  | Nebivolol group | Diltiazem group | Combination group | p-value |
|---|---|---|---|---|
| **Changes in diameter, mm** | 1.44±0.28[a] | 2.04±0.60[b] | 1.30±0.47[a] | <0.001 |
| **Percent changes in diameter, %** | 50.4±8.8[a] | 67.8±12.8[b] | 46.8±12.3[a] | 0.0054 |

Values are presented as number of patients (%) or mean±standard deviation. CAG, coronary angiography; IC, intra-coronary; NTG, nitroglycerin; ACH, acetylcholine.

[a,b] Tukey-Kramer test in linear mixed model were performed after adjusting for the effects of total cholesterol and LDL. Changes and percent changes in diameter is significantly greater in the Diltiazem group than the Nebivolol group or the Combination group.

overall Seattle Angina Questionnaire score from baseline was observed in the overall study population (p = 0.0002). Baseline levels of overall Seattle Angina Questionnaire score between groups, change over time within groups, and difference regarding change over time between 3 groups were all insignificant (Fig 3).

## Blood pressure measurement at baseline and at each visit

Changes in systolic, diastolic and pulse blood pressures during the 12-week follow-up are shown at Table 5 and Fig 4. Details regarding blood pressures at 5 visits are shown in S2 Table. After 12 weeks of study medication, significant systolic blood pressure reduction from baseline was observed in the overall population (p<0.0001). Baseline levels of systolic pressure, diastolic pressure, and pulse pressure between groups, their changes over time within groups, and difference regarding changes over time between 3 groups were all insignificant.

## Inflammatory markers

Changes in inflammatory cytokines during the 12-week follow-up are shown in Table 6. Baseline levels of inflammatory cytokines, including IL-6, TNF-α, hsCRP, sVCAM-1, sICAM-1, and adiponectin, were similar among the three groups. There were no significant differences in inflammatory cytokine levels from baseline during the 12-week follow-up among the three groups.

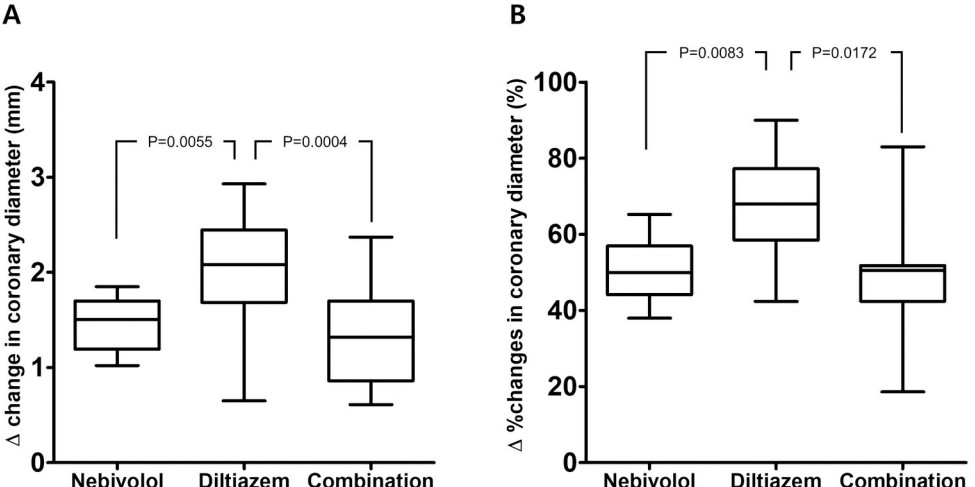

**Fig 2. Vessel diameter changes in coronary artery spasm after intracoronary acetylcholine administration at 12 weeks from baseline among the 3 groups.** Panel (A) denotes changes in diameter. Panel (B) denotes percent changes in diameter.

**Table 4. Changes in Seattle Angina Questionnaire during the 12-week follow-up.**

|  | Seattle Angina Questionnaire overall score | | | |
|---|---|---|---|---|
|  | Baseline | 12-week | Changes | p-value[a] |
| **Nebivolol** | 81.0±9.2 | 86.9±5.0 | 5.9±8.0 | - |
| **Diltiazem** | 80.6±7.9 | 82.5±10.4 | 2.2±9.3 | - |
| **Combination** | 79.5±7.2 | 86.6±6.9 | 7.1±8.1 | - |
| **Overall** | 80.3±7.9 | 85.4±7.8 | 5.2±8.5 | 0.0002 |

Values are presented as number of patients (%) or mean±standard deviation.

[a] p-value was for the effect of time in linear mixed model after adjusting for the effects of sex and LDL. The effect of study group in linear model was not significant (p-value = 0.824).

## Discussion

In this prospective, randomized, double-blind pilot study, we compared the effects of nebivolol and diltiazem on improvements in coronary artery vasospasm, quality of life, and blood pressure in hypertensive patients with documented coronary vasospasm. The major findings of the present study are as follows: (1) significant reduction in coronary vasospasm was observed in all three groups, but the magnitude of reduction was the greatest in the Diltiazem Group; (2) though the reduction in coronary vasospasm was significantly greater in the Diltiazem Group, significant improvement in quality of life based on Seattle Angina Questionnaire after 12 weeks of medication was observed in the overall study population; and (3) modest systolic blood pressure-lowering effects were observed in overall study population.

### Vasoconstrictive effects of beta-blockers in vasospastic angina

Beta-blockers are contraindicated for their use in treating patients with vasospastic angina because they could aggravate coronary spasms by leaving alpha-mediated vasoconstriction [4]. Propranolol has been reported to induce coronary vasospasm [18], and metoprolol and bisoprolol, which are beta-receptor antagonists with beta-1 receptor selectivity, have also been shown to induce coronary vasospasm [19, 20]. Labetalol, which is a selective beta-blocker and a postsynaptic alpha-1 blocking agent, has been reported to induce coronary vasospasm episodes in a patient with aortic dissection by unopposed alpha-adrenergic activity [21]. Moreover, some beta-blockers have been associated with the induction of multi-vessel vasospasm, eventually resulting in ventricular fibrillation [20, 22]. In a Japanese study, patients with post-

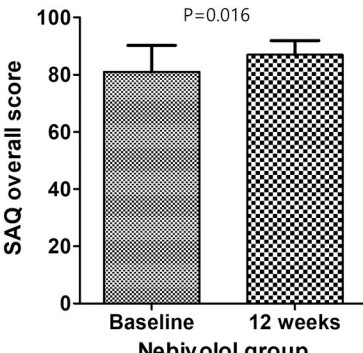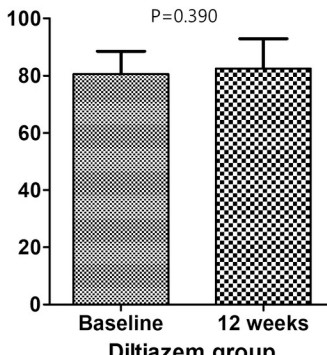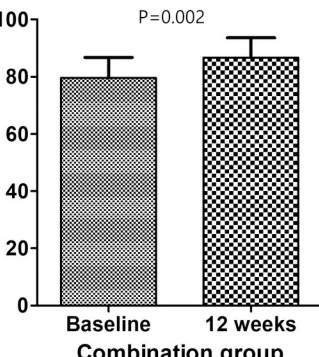

**Fig 3. Changes in quality of life after 12 weeks of random study medication at 12 weeks from baseline among the 3 groups.**
SAQ = Seattle Angina Questionnaire.

**Table 5. Changes in blood pressures during the 12-week follow-up.**

| | Nebivolol group | Diltiazem group | Combination group | Overall population |
|---|---|---|---|---|
| **Changes in SBP** | -8.5±24.6 | -15.4±20.7 | -12.5±15.9 | -12.2±20.1 |
| **Changes in DBP** | -10.4±12.6 | -5.2±10.5 | -12.2±14.1 | -9.5±12.7 |
| **Changes in pulse pressure** | 1.9±17.2 | -10.1±16.6 | -0.3±16.4 | -2.7±17.1 |

Values are presented as number of patients (%) or mean±standard deviation. SBP, systolic blood pressure; DBP, diastolic blood pressure.

myocardial infarction receiving beta-blockers were found to have coronary spasms more frequently than those receiving calcium channel blockers [23]. As a result, Takagi et al. reported a risk score for vasospastic angina, which included the use of beta-blockers as one of the predictive factors for major adverse cardiac events [24]. Therefore, the use of beta-blockers in patients with vasospastic angina have been avoided or limited to special conditions such as concomitant significant stenosis of the coronary artery, left ventricular dysfunction, hypertrophic cardiomyopathy, and myocardial bridging [25].

### Nebivolol as a third-generation nitric oxide-releasing beta-blocker

Nebivolol, the third-generation beta-blocker, is a selective beta-1 blocker with vasodilating activity via the release of nitric oxide [13]. It has been shown in an animal study that nebivolol could induce canine coronary artery vasodilation, which was induced by the release of endothelium-derived relaxing factors [26]. Moreover, nebivolol attenuated cerebral vasospasm by increasing nitric oxide and decreasing oxidative stress in a rat study [27]. Cockcroft et al. also reported that nebivolol dilated human forearm vasculature through its L-arginine/nitric oxide pathway [28]. Though the vasodilating effects of nebivolol on human coronary artery have not been clearly addressed so far, one *in vitro* study has shown the vasorelaxation of human coronary micro vessels facilitated by nebivolol [14]. In our study, the Nebivolol and Low-dose Combination Groups, although not as potent as the Diltiazem Group, showed modest vasodilating effects in the coronary angiography follow-ups. To the best of our knowledge, this is the first study to demonstrate the coronary vasodilation effect of nebivolol in an *in vivo* setting. Sen et al. previously reported that patients with syndrome X have shown a significant improvement in angina after being treated with nebivolol compared to metoprolol [29]. Our study has also demonstrated that, together with diltiazem, nebivolol improved the quality of life. The anti-anxiolytic effect of beta-blockers could have influenced the quality of life in our study

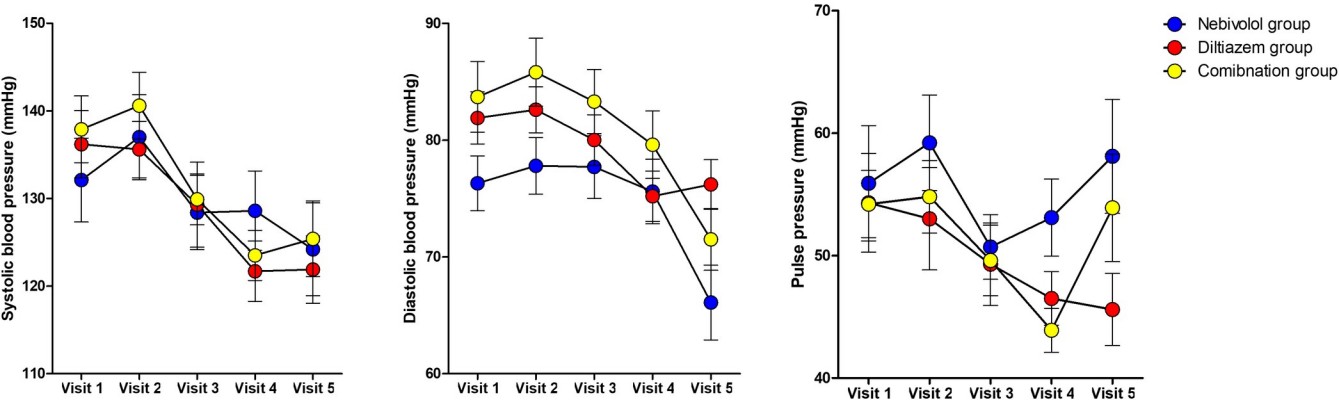

**Fig 4. Changes in blood pressure after 12 weeks of study medication at 12 weeks from baseline among the 3 groups.**

**Table 6. Changes in inflammatory cytokines during the 12-week follow-up.**

| Baseline | Nebivolol group (n = 12) | Diltiazem group (n = 12) | Combination group (n = 11) | p-value |
|---|---|---|---|---|
| Interleukin-6 | 2.07±0.95 | 1.72±0.68 | 1.78±0.99 | 0.610 |
| TNF-α | 14.44±5.26 | 17.09±6.87 | 19.15±6.35 | 0.222 |
| hsCRP | 3.41±2.39 | 3.72±2.63 | 3.03±1.89 | 0.776 |
| sVCAM-1 | 1193.5±335.8 | 1073.7±343.3 | 1151.3±374.5 | 0.710 |
| sICAM-1 | 461.6±109.8 | 501.6±223.3 | 469.3±111.3 | 0.816 |
| Adiponectin | 7.37±2.57 | 6.77±1.81 | 8.77±1.96 | 0.087 |
| **12 weeks** | **Nebivolol group (n = 10)** | **Diltiazem group (n = 10)** | **Combination group (n = 9)** | **p-value** |
| Interleukin-6 | 1.53±0.82 | 1.80±0.55 | 2.01±0.99 | 0.436 |
| TNF-α | 16.35±5.25 | 16.67±5.42 | 16.73±7.12 | 0.989 |
| hsCRP | 1.95±1.07 | 2.52±1.26 | 2.21±1.08 | 0.547 |
| sVCAM-1 | 1124.6±266.5 | 1057.4±230.3 | 1021.1±228.0 | 0.644 |
| sICAM-1 | 462.7±91.0 | 543.8±96.0 | 466.6±95.6 | 0.118 |
| Adiponectin | 9.24±2.07 | 9.29±2.25 | 10.75±2.48 | 0.279 |
| **Changes in levels of inflammatory markers between V1 and V5** | | | | |
| Delta changes in Interleukin-6 | -0.44±1.48 | 0.07±0.91 | 0.41±1.65 | 0.406 |
| Delta changes in TNF a | 2.14±7.13 | -2.19±8.44 | -3.13±10.30 | 0.373 |
| Delta changes in hsCRP | -1.43±3.10 | -1.07±3.71 | -0.34±1.67 | 0.728 |
| Delta changes in sVCAM-1 | -87.3±405.3 | -11.0±253.5 | -124.8±410.9 | 0.783 |
| Delta changes in sICAM-1 | -15.2±146.2 | 10.8±239.8 | 24.5±124.9 | 0.886 |
| Delta changes in Adiponectin | 1.69±3.61 | 2.22±3.19 | 1.60±2.65 | 0.897 |

Values are presented as number of patients (%) or mean±standard deviation. TNF-α, tumor necrosis factor-α; hsCRP, high-sensitivity C-reactive protein; sVCAM-1, soluble vascular cell adhesion molecule-1; sICAM-1, soluble intercellular adhesion molecule-1.

population. In addition, although the degree of vasospasm reduction in the Nebivolol group was less than in the Diltiazem group, follow-up coronary angiography showed no positive findings in the Nebivolol group, so modest relief of vasospasm may be sufficient to improve angina by vasospasm. A previous study reported that nebivolol may improve endothelial dysfunction, and it seems that this mechanism may have contributed to symptomatic improvement of vasospastic angina [13]. Kayaalti et al. reported that nebivolol resulted in an increased diameter of the brachial artery and decreased circulating high-sensitivity C-reactive protein levels [30]. However, our study did not show any meaningful improvements regarding circulating inflammatory cytokines during the 12-week follow-up in all three groups. The short duration of the study drug administration and relatively small subject number might have influenced the levels of inflammatory markers.

## Study limitations

This study was a pilot study that confirmed the vasodilating effect of nebivolol in an in-vivo setting; therefore, sample calculation was unavailable for the study. The number of participants and follow-up duration was limited. A clinical trial with a larger number of participants based on adequate sample calculation and long-term follow-up is warranted. This study had no placebo group; hence we could not conclude that each treatment modality provided benefit. The trial only allows us to make comparisons among the 3 treatment groups. However, it was unethical to administer only placebo drugs to severe vasospastic angina patients. Therefore, calcium channel blocker, first choice for the treatment of vasospastic angina had been chosen as a standard control group [25]. Finally, data related to changes in heart rate were not collected in present study.

## Conclusions

Both nebivolol and diltiazem showed significant coronary vasospasm reduction effect, but the effect was greater for diltiazem in hypertensive patients with vasospastic angina. Significant improvement in quality of life was observed in overall study population.

## Supporting information

**S1 Table. Details of changes in Seattle Angina Questionnaire during the 12-week follow-up.**
(DOCX)

**S2 Table. Details of changes in blood pressures during the 12-week follow-up.**
(DOCX)

**S1 File. CONSORT 2010 checklist of information to include when reporting a randomised trial**\*.
(DOC)

**S2 File.**
(DOCX)

**S3 File.**
(DOCX)

**S1 Data.**
(XLSX)

## Author Contributions

**Conceptualization:** Hyungdon Kook, Soon Jun Hong, Chang Gyu Park.

**Data curation:** Hyungdon Kook, Soon Jun Hong, Kyung-Sook Yang, Jung-Sun Kim, Chang Gyu Park.

**Formal analysis:** Hyungdon Kook, Kyung-Sook Yang.

**Funding acquisition:** Soon Jun Hong.

**Investigation:** Sunki Lee.

**Methodology:** Kyung-Sook Yang, Sunki Lee.

**Project administration:** Chang Gyu Park.

**Supervision:** Jung-Sun Kim.

**Validation:** Hyungdon Kook, Jung-Sun Kim.

**Visualization:** Hyungdon Kook.

**Writing – original draft:** Hyungdon Kook.

**Writing – review & editing:** Soon Jun Hong, Kyung-Sook Yang.

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
