## [Decision Letter · Decision Letter 0]

1 Jul 2020

PONE-D-20-15573

Comparison of Nebivolol versus Diltiazem in improving coronary artery spasm and quality of life in patients with hypertension and vasospastic angina: A prospective, randomized, double-blind trial

PLOS ONE

Dear Dr. Hong,

Thank you for submitting your manuscript to PLOS ONE. After careful consideration, we feel that it has merit but does not fully meet PLOS ONE’s publication criteria as it currently stands. Therefore, we invite you to submit a revised version of the manuscript that addresses the points raised during the review process.

We look forward to receiving your revised manuscript.

Kind regards,

Yoshihiro Fukumoto

Academic Editor

PLOS ONE

Journal Requirements:

2. Please refer to any post-hoc corrections to correct for multiple comparisons during your statistical analyses and sample size calculations performed prior to participant recruitment. If these were not performed please justify the reasons. Please refer to our statistical reporting guidelines for assistance (https://journals.plos.org/plosone/s/submission-guidelines.#loc-statistical-reporting).

3. Thank you for submitting your clinical trial to PLOS ONE and for providing the name of the registry and the registration number. The information in the registry entry suggests that your trial was registered after patient recruitment began. PLOS ONE strongly encourages authors to register all trials before recruiting the first participant in a study.

1) your reasons for your delay in registering this study (after enrolment of participants started);

2) confirmation that all related trials are registered by stating: “The authors confirm that all ongoing and related trials for this drug/intervention are registered”.

Please also ensure you report the date at which the ethics committee approved the study as well as the complete date range for patient recruitment and follow-up in the Methods section of your manuscript.

Additional Editor Comments (if provided):

Reviewers' comments:

Reviewer's Responses to Questions

**Comments to the Author**

1. Is the manuscript technically sound, and do the data support the conclusions?

Reviewer #1: Yes

Reviewer #2: No

Reviewer #3: Yes

Reviewer #4: Partly

2. Has the statistical analysis been performed appropriately and rigorously? 

Reviewer #1: Yes

Reviewer #2: No

Reviewer #3: Yes

Reviewer #4: No

3. Have the authors made all data underlying the findings in their manuscript fully available?

Reviewer #1: Yes

Reviewer #2: Yes

Reviewer #3: Yes

Reviewer #4: Yes

4. Is the manuscript presented in an intelligible fashion and written in standard English?

Reviewer #1: Yes

Reviewer #2: No

Reviewer #3: Yes

Reviewer #4: Yes

5. Review Comments to the Author

Reviewer #1: In this manuscript, Kook and coworkers performed the multicenter, prospective, randomized, double-blind trial investigated the effect of nebivolol and diltiazem on improvements in coronary spasm, QOL, and blood pressure in 51 hypertensive spasm patients. The authors showed that significant reduction in coronary vasospasm was observed in all three groups, but the magnitude of reduction was the greatest in the Diltiazem Group, and that significant improvement in QOL based on Seattle Angina Questionnaire after 12 weeks of medication was observed in the Nebivolol and Low-dose Combination Groups although the reduction in coronary vasospasm was significantly greater in the Diltiazem Group. This study needs to be improved in several points.

1. Abstract: In conclusion, the authors should only emphasize the primary but not secondary endpoint.

2. How did the authors perform the double-blind prescription to the participants in the present study?

3. How long did the authors set up the washout period of trial medication before ACh provocation test? How was the positive rate of coronary spasm in 3 treatment groups at 12 weeks after randomization?

4. How do the authors explain the reason why the significant improvement in QOL based on Seattle Angina Questionnaire after 12 weeks of medication was observed in the Nebivolol and Low-dose Combination Groups although the reduction in coronary vasospasm was significantly greater in the Diltiazem Group?

5. It is difficult to properly terminate the present study without sample calculation. How did the authors terminate the registration?

Reviewer #2: In this manuscript entitled, "Comparison of Nebivolol versus Diltiazem in improving coronary artery spasm and quality of life in patients with hypertension and vasospastic angina: A prospective, randomized, double-blind trial", the authors conducted a prospective, randomized, double-blind trial comparing the therapeutic effect of oral nebivolol, oral slow-releasing diltiazem, and low-dose combination of those two drugs in the patients with vasospastic angina (VSA). The authors demonstrated that quality of life was improved significantly in the patients of nebivolol group, although the incidence of intracoronary acetylcholine (Ach)-induced coronary spasm was reduced to the greatest degree.

Comments:

This study focused on assessment of therapeutic efficacy of nebivolol, which is a beta-blocker with nitric oxide-releasing effect, in VSA patients. The reviewer congratulates the authors on their hardships. Unfortunately, however, this study has major concerns as shown below.

1. The reviewer wonders why there was a discrepancy between the suppressive effect in Ach-induced coronary spasm and the clinical outcome after 12 weeks of treatment among the three groups. Those results contradicted each other. Any additional comment on these?

2. Each treatment group had very small sample size, only 15 subjects in Nebivolol group, only 16 subjects in Diltiazem group, and only 17 subjects in Low dose combination group, respectively. The reviewer considers that those sample sizes were too small to come up with the right answer. The authors should mention how they calculated the adequate sample size.

Reviewer #3: The purpose of the present study was quite interesting, and would impact on the guidelines for coronary vasospasm. However, there are several major questions to make this study further scientific.

1) The patients diagnosed with vasospasm treated by nebivolol were not treated by calcium channel blockers. Is there any ethical problem accordance with guidelines in your country?

2) I could not find any statistic reason of the sample size in randomized, prospective study. How did you calculate?

3) The reason why QOL was differed between diltiazem and nebivolol group was still vague. You should discuss this aspect more.

4) How about the changes in heart rate in your study? I feel that changes in heart rate would be different in three groups.

Reviewer #4: General Comments: The authors provide their findings from a randomized clinical trial of Nebivolol vs. Diltiazem vs. combination for treatment of coronary vasospasm. There is some merit to the study design, but the statistical analysis is poorly conducted. The Results, especially Tables 2-4, are confusingly organized and do not allow the reader to easily view changes over time. Not that in this review I am primarily focusing on study design and analytic concerns.

Specific Comments:

1. Abstract, Methods: “…were randomly categorized into 3 treatment groups:” Please use more specific language here, something like “were randomly allocated”.

2. Abstract, Results: For the QoL scores, the definitions of the numbers within the parentheses are not clear: are they comparisons over time, between groups, etc.?

3. Introduction: This section is clear and concise.

4. Methods: Very little about how patients and study team members were blinded is mentioned, other than the efforts to make dosages appear similar in number of pills.

5. Methods: There is no description of the process by which patients were allocated between groups, other than it was in a randomized and balanced manner. How were random assignments generated? Who generated it? How was it concealed from PI? Etc.

6. Methods, Statistical Analysis: The authors state that comparisons were made between groups using ANOVA or Kruskal-Wallis, while changes from baseline to 12 weeks were analyzed using paired t-tests. Both sets of analyses are inappropriate and unacceptable, as the authors have a repeated-measure design (baseline and control). Use of ANOVA ignores (i) potential differences in baseline measurements between patients and between groups and (ii) the dependence between measurements over time within individuals. The paired t-tests are unhelpful because they make no comparisons between groups. The authors instead should consider using either the Repeated Measures ANOVA Model or the Linear Mixed Model, though the latter would generally be the preferred approach. This model would need to include (at a minimum) fixed effects for the three-level group indicator, the two-level time indicator, and an interaction effect, and also include either a random subject effect or a correlation term. With this single model, the authors can compare baseline levels between groups, can estimate change over time within groups, and can determine if change over time is significantly different between groups.

7. Methods, Statistical Analysis: The authors mention that since their study drug (nebivolol) has never been directly compared with diliazem, they were unable to calculate sample size from previously published data. While I believe them, this is not sufficient reason for omitting a sample size justification. Prospective studies are often deemed inadmissible without such justification. Without this, the authors will have to make clear throughout the manuscript – including in the title – that their study is a pilot study.

8. Results, Baseline Characteristics of Patients: The authors claim that the baseline characteristics – including sex – are similar between their three groups. Notably omitted from this statement are smoking history, social drinking, and diabetes, the indicators for which are more greatly positive in the Nebivolol group than in the others. The authors need to make these comparisons (significant or not) in this section, since their study provides no sample size justification.

9. Table 2: The results are provided in a fairly confusing manner, as it is difficult to track changes over time, which the is the ultimate study outcome.

10. Results, Coronary angiographic findings at baseline and at the 12-week follow-up, first paragraph, second sentence: “All patients had no significant coronary artery stenosis.” Please use a different word than “significant” as this is confusing in light its statistical meaning and usage.

11. Results, Coronary angiographic findings at baseline and at the 12-week follow-up, first paragraph, third sentence: “12 weeks of random study medications…” what do the authors mean by “random study medications?”

12. Results, Coronary angiographic findings at baseline and at the 12-week follow-up: The authors only provide the overall p-value for the global ANOVA test, which tells us whether there is or is not statistical evidence for a difference somewhere. They have omitted the important step of multiple comparisons, wherein they identify between which of the three groups a difference exists. Of course, this would mean that a significance level adjustment would need to be included, with a corresponding write-up in the Statistical Analysis section.

13. Table 3: Like Table 2, it is difficult to track change over time for any of these measures, which is the essential point of these secondary analyses.

14. Results, Changes in quality of life during 12-week follow-up, first paragraph, third sentence: We again have the phrase “12 weeks of random study medication.” Please clarify what “random study medication” means over 12 weeks.

15. Results, Changes in quality of life during 12-week follow-up: The authors either did not adjust the significance levels in the tests of change over time in each group for multiple comparison, or they did not report that they did so.

16. Line 267 (blood pressure): “Random study medication” please explain what this means.

17. Lines 268-269 (blood pressure analyses): These p-values were not adjusted for multiple comparisons.

18. Lines 270-273: Same with DBP analyses, need to adjust p-values for multiple comparisons.

19. Discussion, Study Limitations: The authors state their study is explorative. Since they did not power their study for any particular hypothesis, I would agree and suggest that they bill this study as a pilot, including in the title.

6. PLOS authors have the option to publish the peer review history of their article (what does this mean?). If published, this will include your full peer review and any attached files.

Reviewer #1: No

Reviewer #2: No

Reviewer #3: **Yes: **TAKUYA KISHI

Reviewer #4: No

---

## [Author Response · Author response to Decision Letter 0]

13 Aug 2020

Ref: Manuscript Number PONE-D-20-15573

Comparison of Nebivolol versus Diltiazem in improving coronary artery spasm and quality of life in patients with hypertension and vasospastic angina: A prospective, randomized, double-blind trial

Answers to the editor’s and reviewers’ comments 

We thank the editor and the reviewers for their thoughtful and constructive comments on our study. Several issues were raised by the editor and reviewers, and we have addressed these issues point-by-point and enclosed our comments. Indeed, we realized that the comments were insightful, constructive, and helpful in strengthening our manuscript. We have used blue color font to indicate the revised portions of our manuscript for the reviewers. We hope that the editor and the reviewers would be satisfied with our responses and find the revised manuscript suitable for publication in PLOS ONE.

Responses to Editor:

Comment #1. 

Reply to Comment #1:

Thank you for your comments. 

Our manuscript has been properly modified to meet PLOS ONE's style requirements.

Comment #2. 

Please refer to any post-hoc corrections to correct for multiple comparisons during your statistical analyses and sample size calculations performed prior to participant recruitment. If these were not performed please justify the reasons.

Reply to Comment #2:

Thank you for your comments. 

After in-depth consultation with statistician (Kyung-Sook Yang, PhD, Department of Biostatistics, Korea University College of Medicine, Seoul, Korea), all authors and statistician have agreed that the reviewer's point was valid, and re-analysis using a linear mixed model was performed according to the recommendations.

The changes in coronary spasm corresponding to the primary endpoint and differences between groups were the same as in the previous analysis. On the other hand, the SAQ score corresponding to the secondary endpoint appeared to show significant improvement after study medication in overall study population, but baseline levels of overall Seattle Angina Questionnaire score between groups, change over time within groups, and difference regarding change over time between 3 groups were all insignificant.

All data analysis results using the linear mixed model was uploaded separately.

In our study, the sample calculation was impossible because this study was a pilot study dealing with a topic that had never been studied previously. Therefore, patients were recruited for a fixed period of time (January 2018 to March 2019) and registration was then terminated. To clarify that this study is a pilot study, the title of this study has been modified. We additionally stated in the limitation section that sample calculation was not possible because this study was a pilot study.

We revised the Title, Introduction, Methods, and the Discussion section in the manuscript accordingly. 

Title

Comparison of Nebivolol versus Diltiazem in improving coronary artery spasm and quality of life in patients with hypertension and vasospastic angina: A prospective, randomized, double-blind pilot study

Page 6, Line 81-84

Therefore, we performed a prospective, randomized, double-blind pilot study to compare nebivolol versus diltiazem in improving coronary artery spasm and quality of life in hypertensive patients with vasospastic angina during a 12-week follow-up.

Page 12-13, Line 217-229

The baseline demographics comparisons between three groups were performed by using analysis of variance (ANOVA) for normal distribution or Kruskal-Wallis test for non-normal distribution. When the result of ANOVA or Kruskal-Wallis test was significant (p-value<0.05), post-hoc analysis was performed using Bonferroni’s correction, Tukey-Kramer test, and Dunnett’s T3 test as appropriate. The linear mixed model was used for comparing baseline levels of primary endpoint and secondary endpoints between groups, estimating change over time within groups, and determining if change over time is significantly different between groups. The linear mixed model included fixed effects for the three-level group indicator, the two-level time indicator, and an interaction effect, and some covariates. The covariates included for fitting the model were study group, age, sex, primary and secondary endpoints, and baseline demographic variables that was less than p<0.1 when comparing between groups. Changes over time were calculated as the difference in the values at the end of the 12-week treatment and at baseline.

Page 13, Line 231-233

To the best of our knowledge, this was the first pilot study to compare nebivolol versus diltiazem in improving coronary artery spasm; therefore, the sample size could not be calculated from previously published data.

Page 24, Line 335-337

In this prospective, randomized, double-blind pilot study, we compared the effects of nebivolol and diltiazem on improvements in coronary artery vasospasm, quality of life, and blood pressure in hypertensive patients with documented coronary vasospasm.

Page 26, Line 393-396

This study was a pilot study that confirmed the vasodilating effect of nebivolol in an in-vivo setting; therefore, sample calculation was unavailable for the study. The number of participants and follow-up duration was limited. A clinical trial with a larger number of participants based on adequate sample calculation and long-term follow-up is warranted.

Comment #3. 

1) your reasons for your delay in registering this study (after enrolment of participants started);

2) confirmation that all related trials are registered by stating: “The authors confirm that all ongoing and related trials for this drug/intervention are registered”.

Reply to Comment #3:

Thank you for your question. There was a delay in registering this study after enrolment of participant started due to funding issues which have nothing to do with the protocol itself.

We have added in the Method section that “The authors confirm that all ongoing and related trials for this drug/intervention are registered.”

Kept these comments in mind, we revised the Methods section in the manuscript accordingly. 

Page 10, Line 158-160

There was a delay in registering this study after enrolment of participant started due to funding issues which have nothing to do with the protocol itself. The authors confirm that all ongoing and related trials for these drugs are registered.

Comment #4. 

Please also ensure you report the date at which the ethics committee approved the study as well as the complete date range for patient recruitment and follow-up in the Methods section of your manuscript.

Reply to Comment #4:

Thank you for your comments. 

We specified the IRB approval date, the complete date range for patient recruitment and follow-up in the Methods section.

Kept these comments in mind, we revised the Methods section in the manuscript accordingly. 

Page 9, Line 149-153

This study was approved by the Korea University Hospital Institute Review Board (IRB No. 2015AN0296) from January 1, 2018 to December 31, 2019, and written informed consent was obtained from all participants or their legal representatives. Complete date range for patient recruitment was from January 2018 to March 2019. Follow-up duration for this study was 12 weeks.

 

Responses to Reviewer #1:

Comment #1. 

Abstract: In conclusion, the authors should only emphasize the primary but not secondary endpoint.

Reply to Comment #1:

Thank you for your critical comment that we should address. 

As the reviewer’s comment, we deleted the sentence for the secondary endpoint.

Kept this comment in mind, we revised the Abstract section in the manuscript accordingly. 

Page 4, Line 48-49

Both nebivolol and diltiazem showed significant coronary vasospasm reduction effect, but the effect was greater for diltiazem.

Comment #2. 

How did the authors perform the double-blind prescription to the participants in the present study?

Reply to Comment #2:

Thank you for your valuable comments we should address. 

As the reviewer pointed out, we further described in more detail regarding the double-blind prescription process in the paper.

The nebivolol group was given a placebo with the same color, shape, and size as diltiazem, and the diltiazem group was additionally prescribed a placebo with the same color, shape, and size as nebivolol. The low-dose combination group was prescribed by preparing a half-dose drug in the same size as the original dose. Each study drugs were packaged in identical containers.

The attending physician, study subject, personnel at investigative sites, and all study staff and investigators, other than the safety monitoring board were blinded to treatment assignment.

Kept these comments in mind, we revised the Methods section in the manuscript accordingly. 

Page 9, Line 137-143

The Nebivolol group was additionally given a placebo with the same color, shape, and size as diltiazem, and the Diltiazem group was additionally given a placebo with the same color, shape, and size as nebivolol. The Low-dose combination group was given by preparing a half-dose drug in the same size as the original dose. These study drugs were packaged in identical containers. The attending physician, study subject, personnel at investigative sites, and all study staff and investigators, other than the safety monitoring board were blinded to treatment assignment via a locked web-based system.

Comment #3. 

How long did the authors set up the washout period of trial medication before ACh provocation test? How was the positive rate of coronary spasm in 3 treatment groups at 12 weeks after randomization?

Reply to Comment #3:

Thank you for your valid comments that we should address. 

Suitable study subjects who were taking ACE inhibitors, angiotensin blockers, beta blockers, calcium channel blockers, and diuretics other than indapamide were able to participate in the study after a wash-out period of at least 2 weeks. 

None of the patients who completed follow-up coronary angiography at 12 weeks after randomization had more than 75% of vasospasm.

Kept these comments in mind, we revised the Methods section and the Result section in the manuscript accordingly. 

Page 7, Line 98-101

(4) study subjects who were taking drugs that can affect the evaluation of the study drug efficacy (beta-blockers, calcium channel antagonists, nicorandil, and nitrates), but these subjects were allowed to participate after a wash-out period of at least 2 weeks;

Page 16-17, Line 271-273

None of the patients who completed follow-up coronary angiography at 12 weeks after randomization had more than 75% of vasospasm.

Comment #4. 

How do the authors explain the reason why the significant improvement in QOL based on Seattle Angina Questionnaire after 12 weeks of medication was observed in the Nebivolol and Low-dose Combination Groups although the reduction in coronary vasospasm was significantly greater in the Diltiazem Group?

Reply to Comment #4:

We greatly appreciate the reviewer’s comment. 

Although the mechanism cannot be accurately described, we estimate that nebivolol's beta blocker-specific anxiolytic effect, like propranolol, may have affected QOL.

In addition, although the degree of vasospasm reduction in the Nebivolol group was less than in the Diltiazem group, follow-up coronary angiography showed no positive findings in the Nebivolol group, so modest relief of vasospasm may be sufficient to improve angina by vasospasm.

Tzemos N et al. reports that nebivolol may improve endothelial dysfunction in previous studies, and it seems that this mechanism may have contributed to symptomatic improvement of vasospastic angina.

Kept these comments in mind, we revised the Discussion section in the manuscript accordingly. 

Page 25, Line 377-385

Our study has also demonstrated that, together with diltiazem, nebivolol improved the quality of life. The anti-anxiolytic effect of beta-blockers could have influenced the quality of life in our study population. In addition, although the degree of vasospasm reduction in the Nebivolol group was less than in the Diltiazem group, follow-up coronary angiography showed no positive findings in the Nebivolol group, so modest relief of vasospasm may be sufficient to improve angina by vasospasm. A previous study reported that nebivolol may improve endothelial dysfunction, and it seems that this mechanism may have contributed to symptomatic improvement of vasospastic angina.

Comment #5. 

It is difficult to properly terminate the present study without sample calculation. How did the authors terminate the registration?

Reply to Comment #5:

Thank you very much for your valid comments we should address. 

As the reviewer pointed out, the sample calculation was impossible because this study was a pilot study dealing with a topic that had not been studied previously. Therefore, patients were recruited for a fixed period of time (January 2018 to March 2019) and registration was then terminated. To clarify that this study is a pilot study, the title of this study has been modified. We additionally stated in the limitation section that sample calculation was impossible because this study was a pilot study.

Kept these comments in mind, we revised the Title, Methods section and the Discussion section in the manuscript accordingly. 

Title

Comparison of Nebivolol versus Diltiazem in improving coronary artery spasm and quality of life in patients with hypertension and vasospastic angina: A prospective, randomized, double-blind pilot study

Page 9, Line 152-153

Complete date range for patient recruitment was from January 2018 to March 2019.

Page 26, Line 393-396

This study was a pilot study that confirmed the vasodilating effect of nebivolol in an in-vivo setting; therefore, sample calculation was unavailable for the study. The number of participants and follow-up duration was limited. A clinical trial with a larger number of participants based on adequate sample calculation and long-term follow-up is warranted.

 

Responses to Reviewer #2:

Comment #1. 

The reviewer wonders why there was a discrepancy between the suppressive effect in Ach-induced coronary spasm and the clinical outcome after 12 weeks of treatment among the three groups. Those results contradicted each other. Any additional comment on these?

Reply to Comment #1:

Thank you for your insightful comments that we should address. 

Although the mechanism cannot be accurately described, we estimate that nebivolol's beta blocker-specific anxiolytic effect, like propranolol, may have affected clinical outcome (quality of life based on Seattle Angina Questionnaire).

In addition, although the degree of vasospasm reduction in the Nebivolol group was less than in the Diltiazem group, follow-up coronary angiography showed no positive findings in the Nebivolol group, so modest relief of vasospasm may be sufficient to improve angina by vasospasm.

Tzemos N et al. reports that nebivolol may improve endothelial dysfunction in previous studies, and it seems that this mechanism may have contributed to symptomatic improvement of vasospastic angina.

Kept these comments in mind, we revised the Discussion section in the manuscript accordingly. 

Page 25-26, Line 377-385

Our study has also demonstrated that, together with diltiazem, nebivolol improved the quality of life. The anti-anxiolytic effect of beta-blockers could have influenced the quality of life in our study population. In addition, although the degree of vasospasm reduction in the Nebivolol group was less than in the Diltiazem group, follow-up coronary angiography showed no positive findings in the Nebivolol group, so modest relief of vasospasm may be sufficient to improve angina by vasospasm. A previous study reported that nebivolol may improve endothelial dysfunction, and it seems that this mechanism may have contributed to symptomatic improvement of vasospastic angina.

Comment #2. 

Each treatment group had very small sample size, only 15 subjects in Nebivolol group, only 16 subjects in Diltiazem group, and only 17 subjects in Low dose combination group, respectively. The reviewer considers that those sample sizes were too small to come up with the right answer. The authors should mention how they calculated the adequate sample size.

Reply to Comment #2:

Thank you for your valuable comments and we sincerely sympathize with your opinions. 

As the reviewer pointed out, the sample calculation was impossible because this study was a pilot study dealing with a topic that had not been studied previously. Therefore, patients were recruited for a fixed period of time (January 2018 to March 2019) and registration was then terminated. To clarify that this study is a pilot study, the title of this study has been modified. We additionally stated in the limitation section that sample calculation was impossible because this study was a pilot study.

Kept these comments in mind, we revised the Title, Methods section and the Discussion section in the manuscript accordingly. 

Title

Comparison of Nebivolol versus Diltiazem in improving coronary artery spasm and quality of life in patients with hypertension and vasospastic angina: A prospective, randomized, double-blind pilot study

Page 9, Line 152-153

Complete date range for patient recruitment was from January 2018 to March 2019.

Page 26, Line 393-396

This study was a pilot study that confirmed the vasodilating effect of nebivolol in an in-vivo setting; therefore, sample calculation was unavailable for the study. The number of participants and follow-up duration was limited. A clinical trial with a larger number of participants based on adequate sample calculation and long-term follow-up is warranted.

 

Responses to Reviewer #3:

Comment #1. 

The patients diagnosed with vasospasm treated by nebivolol were not treated by calcium channel blockers. Is there any ethical problem accordance with guidelines in your country?

Reply to Comment #1:

Thank you for your critical comments that we should address.

To the best of our knowledge, there is no clear guideline for medical therapy of vasospastic angina other than the guideline for vasospastic angina published in the Circulation Journal in 2014. Although calcium channel blocker is recommended as the first choice in vasospastic angina in the guideline, it is also possible to use alternative medications such as nitrate that can be metabolized to NO in the body, instead of calcium channel blocker. Therefore, prescription of nebivolol alone, which has NO releasing effect, did not cause ethical problems in our country.

Comment #2. 

I could not find any statistic reason of the sample size in randomized, prospective study. How did you calculate?

Reply to Comment #2:

Thank you for your insightful comments that we should address. 

In our study, the sample calculation was impossible because this study was a pilot study dealing with a topic that had not been studied previously. Therefore, patients were recruited for a fixed period of time (January 2018 to March 2019) and registration was then terminated. To clarify that this study is a pilot study, the title of this study has been modified. We additionally stated in the limitation section that sample calculation was impossible because this study was a pilot study.

Kept these comments in mind, we revised the Title, Methods section and the Discussion section in the manuscript accordingly. 

Title

Comparison of Nebivolol versus Diltiazem in improving coronary artery spasm and quality of life in patients with hypertension and vasospastic angina: A prospective, randomized, double-blind pilot study

Page 9, Line 152-153

Complete date range for patient recruitment was from January 2018 to March 2019.

Page 26, Line 393-396

This study was a pilot study that confirmed the vasodilating effect of nebivolol in an in-vivo setting; therefore, sample calculation was unavailable for the study. The number of participants and follow-up duration was limited. A clinical trial with a larger number of participants based on adequate sample calculation and long-term follow-up is warranted.

Comment #3. 

The reason why QOL was differed between diltiazem and nebivolol group was still vague. You should discuss this aspect more.

Reply to Comment #3:

Thank you for your valuable comments and we totally agreed with your opinion. 

Although the mechanism cannot be accurately described through this study, we estimate that nebivolol's beta blocker-specific anxiolytic effect, like propranolol, may have affected clinical outcome (quality of life based on Seattle Angina Questionnaire).

In addition, although the degree of vasospasm reduction in the Nebivolol group was less than in the Diltiazem group, follow-up coronary angiography with vasospasm provocation test showed no positive findings in the Nebivolol group, so modest relief of vasospasm may be sufficient to improve angina by vasospasm.

Tzemos N et al. reports that nebivolol may improve endothelial dysfunction in previous studies, and it seems that this mechanism may have contributed to symptomatic improvement of vasospastic angina.

However, following other reviewer’s comment regarding statistical analysis methods, we re-analyze our primary and secondary endpoints using a linear mixed model. As a result, changes in each group of coronary spasm corresponding to the primary endpoint and differences between groups were the same as in the previous analysis. On the other hand, the Seattle Angina Questionnaire score corresponding to the secondary endpoint appeared to show significant improvement after study medication in overall study population, but baseline levels of overall Seattle Angina Questionnaire score between groups, change over time within groups, and difference regarding change over time between 3 groups were all insignificant.

All data analysis results using the linear mixed model was uploaded separately.

Based on these re-analysis results, we have revised the Abstract, Result, Discussion sections, and Tables in the manuscript accordingly.

Page 3, Line 39-46

however, the improvement in percent changes in coronary artery spasm was greatest in the Diltiazem Group (50.4±8.8% vs. 67.8±12.8% vs. 46.8±12.3%, Nebivolol Group vs. Diltiazem Group p=0.008; Nebivolol Group vs. Low-dose Combination Group p=0.999; Diltiazem Group vs. Low-dose Combination Group p=0.017). The overall Seattle Angina Questionnaire scores were significantly elevated at 12 weeks compared to the baseline in entire study population. There were no significant differences between the three groups in the overall Seattle Angina Questionnaire score changes and blood pressure changes.

Page 16, Line 257-262

In baseline coronary angiography, significant vasospasm after intracoronary acetylcholine administration in linear mixed model was observed in all three groups without statistical differences (percent changes in diameter in the Nebivolol Group vs. Diltiazem Group vs. Low-dose Combination Group: -87.9±4.9% vs. -86.4±6.3% vs. -85.5±5.7%, Nebivolol Group vs. Diltiazem Group p=0.977; Nebivolol Group vs. Low-dose Combination Group p=0.838; Diltiazem Group vs. Low-dose Combination Group p=0.997).

Page 16, Line 266-271

but the magnitude of improvement in vasospasm was the greatest in the Diltiazem Group (percent changes in diameter from the baseline provocation test for Nebivolol Group vs. Diltiazem Group vs. Low-dose Combination Group: 50.4±8.8% vs. 67.8±12.8% vs. 46.8±12.3%, Nebivolol Group vs. Diltiazem Group p=0.008; Nebivolol Group vs. Low-dose Combination Group p=0.999; Diltiazem Group vs. Low-dose Combination Group p=0.017).

Page 20, Line 292-296

After 12 weeks of study medications, a significant improvement in quality of life based on the overall Seattle Angina Questionnaire score from baseline was observed in the overall study population. Baseline levels of overall Seattle Angina Questionnaire score between groups, change over time within groups, and difference regarding change over time between 3 groups were all insignificant.

Page 24, Line 339-342

(2) though the reduction in coronary vasospasm was significantly greater in the Diltiazem Group, significant improvement in quality of life based on Seattle Angina Questionnaire after 12 weeks of medication was observed in the overall study population;

Page 25-26, Line 377-385

Our study has also demonstrated that, together with diltiazem, nebivolol improved the quality of life. The anti-anxiolytic effect of beta-blockers could have influenced the quality of life in our study population. In addition, although the degree of vasospasm reduction in the Nebivolol group was less than in the Diltiazem group, follow-up coronary angiography showed no positive findings in the Nebivolol group, so modest relief of vasospasm may be sufficient to improve angina by vasospasm. A previous study reported that nebivolol may improve endothelial dysfunction, and it seems that this mechanism may have contributed to symptomatic improvement of vasospastic angina.

Page 27, Line 405-407

Both nebivolol and diltiazem showed significant coronary vasospasm reduction effect, but the effect was greater for diltiazem in hypertensive patients with vasospastic angina. Significant improvement in quality of life was observed in overall study population.

  

Table 2. Coronary angiographic changes during the 12-week follow-up

 Baseline CAG 12-week follow-up CAG p-valuea

 Vessel diameter, mm Vasospasm Vessel diameter, mm Vasospasm 

 IC NTG IC ACH mm % IC NTG IC ACH mm % 

Nebivolol (n=15) 2.98±0.54 0.37±0.17 -2.61±0.44 -87.9±4.9 (n=14) 3.06±0.42 1.90±0.26 -1.16±0.28 -37.5±6.4 <0.001

Diltiazem (n=16) 3.07±0.49 0.42±0.22 -2.65±0.44 -86.4±6.3 (n=13) 3.14±0.38 2.56±0.49 -0.58±0.28 -18.8±9.4 <0.001

Combination (n=17) 2.93±0.68 0.43±0.20 -2.50±0.60 -85.5±5.7 (n=15) 2.87±0.57 1.75±0.42 -1.13±0.37 -38.9±8.6 <0.001

Values are presented as number of patients (%) or mean±standard deviation. CAG, coronary angiography; IC, intra-coronary; NTG, nitroglycerin; ACH, acetylcholine. ap-value was obtained by Tukey-Kramer test for the interaction effect between time and group for diameter percent change in linear mixed model. 

Table 3. Vessel diameter changes between baseline and 12-week follow-up

 Nebivolol group Diltiazem group Combination group p-value

Changes in diameter, mm 1.44±0.28 2.04±0.60 1.30±0.47 <0.001

Percent changes in diameter, % 50.4±8.8a 67.8±12.8b 46.8±12.3a 0.0054

Values are presented as number of patients (%) or mean±standard deviation. CAG, coronary angiography; IC, intra-coronary; NTG, nitroglycerin; ACH, acetylcholine. a,b Tukey-Kramer test in linear mixed model were performed after adjusting for the effects of total cholesterol and LDL. Changes and percent changes in diameter is significantly greater in the Diltiazem group than the Nebivolol group or the Combination group.

 

Table 4. Changes in Seattle Angina Questionnaire during the 12-week follow-up

 Seattle Angina Questionnaire overall score 

 Baseline 12-week Changes p-valuea

Nebivolol 81.0±9.2 86.9±5.0 5.9±8.0 -

Diltiazem 80.6±7.9 82.5±10.4 2.2±9.3 -

Combination 79.5±7.2 86.6±6.9 7.1±8.1 -

Overall 80.3±7.9 85.4±7.8 5.2±8.5 0.0002

Values are presented as number of patients (%) or mean±standard deviation. a p-value was for the effect of time in linear mixed model after adjusting for the effects of sex and LDL. The effect of study group in linear model was not significant (p-value=0.824).

 

Comment #4. 

How about the changes in heart rate in your study? I feel that changes in heart rate would be different in three groups.

Reply to Comment #4:

Thank you for your valid comments that we should address. 

Unfortunately, in this study, since data on heart rate was not collected, the difference in heart rate changes could not be compared. However, since both nebivolol and diltiazem are drugs with a negative chronotropic effect, it is estimated that all groups had a reduced heart rate after 12 weeks of drug treatment.

In the limitation section, it was further stated that no data related to heart rate were collected in this study.

Kept these comments in mind, we revised the Discussion section in the manuscript accordingly.

Page 27, Line 401-402

Finally, data related to changes in heart rate were not collected in present study.

 

Responses to Reviewer #4:

Comment #1. 

Abstract, Methods: “…were randomly categorized into 3 treatment groups:” Please use more specific language here, something like “were randomly allocated”.

Reply to Comment #1:

Thank you for your comments that we should address. The sentence was amended according to the reviewer's recommendation.

Kept these comments in mind, we revised the Abstract and Method section in the manuscript accordingly. 

Page 3, Line 31-32 

Fifty-one hypertensive patients with documented coronary vasospasm were randomly allocated into 3 treatment groups:

Page 8, Line 128-130

A total of 51 hypertensive patients with documented coronary vasospasm were equally and randomly allocated into 3 treatment groups:

Comment #2. 

Abstract, Results: For the QoL scores, the definitions of the numbers within the parentheses are not clear: are they comparisons over time, between groups, etc.?

Reply to Comment #2:

Thank you for your insightful comments that we should address.

Following reviewer’s comment regarding statistical analysis methods, we re-analyze our primary and secondary endpoints using a linear mixed model. As a result, the SAQ score corresponding to the secondary endpoint appeared to show significant improvement after study medication in overall study population, but baseline levels of overall Seattle Angina Questionnaire score between groups, change over time within groups, and difference regarding change over time between 3 groups were all insignificant.

Based on these re-analysis results, we have revised the Abstract and Result sections in the manuscript accordingly.

Page 3, Line 43-46 

The overall Seattle Angina Questionnaire scores were significantly elevated at 12 weeks compared to the baseline in entire study population. There were no significant differences between the three groups in the overall Seattle Angina Questionnaire score changes and blood pressure changes.

Page 20, Line 291-296 

After 12 weeks of study medications, a significant improvement in quality of life based on the overall Seattle Angina Questionnaire score from baseline was observed in the overall study population. Baseline levels of overall Seattle Angina Questionnaire score between groups, change over time within groups, and difference regarding change over time between 3 groups were all insignificant.

Comment #3. 

Methods: Very little about how patients and study team members were blinded is mentioned, other than the efforts to make dosages appear similar in number of pills.

Reply to Comment #3:

Thank you for your valuable comments. 

As the reviewer pointed out, we further described in more detail regarding the double-blind prescription process in the paper.

The nebivolol group was given a placebo with the same color, shape, and size as diltiazem, and the diltiazem group was additionally prescribed a placebo with the same color, shape, and size as nebivolol. The low-dose combination group was prescribed by preparing a half-dose drug in the same size as the original dose. Each study drugs were packaged in identical containers.

The attending physician, study subject, personnel at investigative sites; and all study staff and investigators, other than the safety monitoring board were blinded to treatment assignment.

Kept these comments in mind, we revised the Methods section in the manuscript accordingly. 

Page 9, Line 137-143

The Nebivolol group was additionally given a placebo with the same color, shape, and size as diltiazem, and the Diltiazem group was additionally given a placebo with the same color, shape, and size as nebivolol. The Low-dose combination group was given by preparing a half-dose drug in the same size as the original dose. These study drugs were packaged in identical containers. The attending physician, study subject, personnel at investigative sites, and all study staff and investigators, other than the safety monitoring board were blinded to treatment assignment via a locked web-based system.

Comment #4. 

Methods: There is no description of the process by which patients were allocated between groups, other than it was in a randomized and balanced manner. How were random assignments generated? Who generated it? How was it concealed from PI? Etc.

Reply to Comment #4:

Thank you for your valid comments that we should address. 

As the reviewer pointed out, the manuscript was reinforced with further information on the process assigned to each group of patients, random assignment generation, and conceal methods.

The attending physician, study subject, personnel at investigative sites, and all study staff and investigators, other than the safety monitoring board were blinded to treatment assignment via a locked web-based system.

Patients who had complained of chest pain with a left anterior descending coronary artery spasm > 75% after the acetylcholine provocation test and had agreed to participate in the study were centrally randomly assigned to a computer-generated random assignment number in a 1:1:1 ratio, according to the randomization table. Randomization was balanced with a block randomization method. The web-based computer-generated randomization table was developed by an external programmer who was not otherwise involved in the trial. 

Kept these comments in mind, we revised the Methods section in the manuscript accordingly.

Page 9, Line 141-149

The attending physician, study subject, personnel at investigative sites, and all study staff and investigators, other than the safety monitoring board were blinded to treatment assignment via a locked web-based system. Patients who had complained of chest pain with a left anterior descending coronary artery spasm > 75% after the acetylcholine provocation test and had agreed to participate in the study were centrally randomly assigned to a computer-generated random assignment number in a 1:1:1 ratio, according to the randomization table. Randomization was balanced with a block randomization method. The web-based computer-generated randomization table was developed by an external programmer who was not otherwise involved in the trial.

Comment #5. 

Methods, Statistical Analysis: The authors state that comparisons were made between groups using ANOVA or Kruskal-Wallis, while changes from baseline to 12 weeks were analyzed using paired t-tests. Both sets of analyses are inappropriate and unacceptable, as the authors have a repeated-measure design (baseline and control). Use of ANOVA ignores (i) potential differences in baseline measurements between patients and between groups and (ii) the dependence between measurements over time within individuals. The paired t-tests are unhelpful because they make no comparisons between groups. The authors instead should consider using either the Repeated Measures ANOVA Model or the Linear Mixed Model, though the latter would generally be the preferred approach. This model would need to include (at a minimum) fixed effects for the three-level group indicator, the two-level time indicator, and an interaction effect, and also include either a random subject effect or a correlation term. With this single model, the authors can compare baseline levels between groups, can estimate change over time within groups, and can determine if change over time is significantly different between groups.

Reply to Comment #5:

Thank you for your critical comments we should address. After in-depth consultation with statistician (Kyung-Sook Yang, PhD, Department of Biostatistics, Korea University College of Medicine, Seoul, Korea) with reviewer’s comment, all authors and statisticians agreed that the reviewer's point was valid, and re-analysis using a linear-mixed model was performed according to recommendations.

As a result, changes in each group of coronary spasm corresponding to the primary endpoint and differences between groups were the same as in the previous analysis. On the other hand, the SAQ score corresponding to the secondary endpoint appeared to show significant improvement after study medication in overall study population, but baseline levels of overall Seattle Angina Questionnaire score between groups, change over time within groups, and difference regarding change over time between 3 groups were all insignificant.

All data analysis results using the linear mixed model was uploaded separately.

Kept these comments in mind, we revised the Methods section in the manuscript accordingly.

Page 12, Line 217-229

The baseline demographics comparisons between three groups were performed by using analysis of variance (ANOVA) for normal distribution or Kruskal-Wallis test for non-normal distribution. When the result of ANOVA or Kruskal-Wallis test was significant (p-value<0.05), post-hoc analysis was performed using Bonferroni’s correction, Tukey-Kramer test, and Dunnett’s T3 test as appropriate. The linear mixed model was used for comparing baseline levels of primary endpoint and secondary endpoints between groups, estimating change over time within groups, and determining if change over time is significantly different between groups. The linear mixed model included fixed effects for the three-level group indicator, the two-level time indicator, and an interaction effect, and some covariates. The covariates included for fitting the model were study group, age, sex, primary and secondary endpoints, and baseline demographic variables that was less than p<0.1 when comparing between groups. Changes over time were calculated as the difference in the values at the end of the 12-week treatment and at baseline.

Comment #6. 

Methods, Statistical Analysis: The authors mention that since their study drug (nebivolol) has never been directly compared with diliazem, they were unable to calculate sample size from previously published data. While I believe them, this is not sufficient reason for omitting a sample size justification. Prospective studies are often deemed inadmissible without such justification. Without this, the authors will have to make clear throughout the manuscript – including in the title – that their study is a pilot study.

Reply to Comment #6:

Thank you for your valuable comments and we agreed with your opinions. In our study, the sample calculation was impossible because this study was a pilot study dealing with a topic that had not been studied previously. Therefore, patients were recruited for a fixed period of time (January 2018 to March 2019) and registration was then terminated. To clarify that this study is a pilot study, the title of this study has been modified. We additionally stated in the limitation section that sample calculation was impossible because this study was a pilot study.

Kept these comments in mind, we revised the Title, Introduction, Methods, and the Discussion section in the manuscript accordingly. 

Title

Comparison of Nebivolol versus Diltiazem in improving coronary artery spasm and quality of life in patients with hypertension and vasospastic angina: A prospective, randomized, double-blind pilot study

Page 6, Line 81-84

Therefore, we performed a prospective, randomized, double-blind pilot study to compare nebivolol versus diltiazem in improving coronary artery spasm and quality of life in hypertensive patients with vasospastic angina during a 12-week follow-up.

Page 13, Line 231-233

To the best of our knowledge, this was the first pilot study to compare nebivolol versus diltiazem in improving coronary artery spasm; therefore, the sample size could not be calculated from previously published data.

Page 24, Line 335-337

In this prospective, randomized, double-blind pilot study, we compared the effects of nebivolol and diltiazem on improvements in coronary artery vasospasm, quality of life, and blood pressure in hypertensive patients with documented coronary vasospasm.

Page 26, Line 393-396

This study was a pilot study that confirmed the vasodilating effect of nebivolol in an in-vivo setting; therefore, sample calculation was unavailable for the study. The number of participants and follow-up duration was limited. A clinical trial with a larger number of participants based on adequate sample calculation and long-term follow-up is warranted.

Comment #7. 

Results, Baseline Characteristics of Patients: The authors claim that the baseline characteristics – including sex – are similar between their three groups. Notably omitted from this statement are smoking history, social drinking, and diabetes, the indicators for which are more greatly positive in the Nebivolol group than in the others. The authors need to make these comparisons (significant or not) in this section, since their study provides no sample size justification.

Reply to Comment #7:

Thank you for your valuable comments and we sincerely sympathize with your opinions. 

When analyzed by ANOVA analysis, the frequency of smoking and diabetes in the Nebivolol group was numerically high but not statistically significant. These results were similar when analyzed through post-hoc analysis. However, the incidence of social drinking tended to be lower in the Combination group than it was in the Nebivolol group (Odds ratio 0.222, 95% CI 0.046-1.083, p=0.076).

Kept these comments in mind, we revised the Result section in the manuscript accordingly. 

Page 14, Line 244-247

The frequency of smoking and diabetes in the Nebivolol group was numerically high but not statistically significant. The frequency of social drinking tended to be lower in the Combination group than it was in the Nebivolol group (Odds ratio 0.222, 95% CI 0.046-1.083, p=0.076).

Comment #8. 

Table 2: The results are provided in a fairly confusing manner, as it is difficult to track changes over time, which the is the ultimate study outcome.

Reply to Comment #8:

Thank you for your valuable comments that we should address. 

The results of the re-analysis using the linear mixed model were reflected in the tables, and table 2 was modified to make it easier to see the track change. In addition, the vessel diameter changes between baseline and 12-week follow-up were separated into Table 3.

Kept these comments in mind, we revised the Table 2 and Table 3 in the manuscript accordingly. 

 

Table 2. Coronary angiographic changes during the 12-week follow-up

 Baseline CAG 12-week follow-up CAG p-valuea

 Vessel diameter, mm Vasospasm Vessel diameter, mm Vasospasm 

 IC NTG IC ACH mm % IC NTG IC ACH mm % 

Nebivolol (n=15) 2.98±0.54 0.37±0.17 -2.61±0.44 -87.9±4.9 (n=14) 3.06±0.42 1.90±0.26 -1.16±0.28 -37.5±6.4 <0.001

Diltiazem (n=16) 3.07±0.49 0.42±0.22 -2.65±0.44 -86.4±6.3 (n=13) 3.14±0.38 2.56±0.49 -0.58±0.28 -18.8±9.4 <0.001

Combination (n=17) 2.93±0.68 0.43±0.20 -2.50±0.60 -85.5±5.7 (n=15) 2.87±0.57 1.75±0.42 -1.13±0.37 -38.9±8.6 <0.001

Values are presented as number of patients (%) or mean±standard deviation. CAG, coronary angiography; IC, intra-coronary; NTG, nitroglycerin; ACH, acetylcholine. ap-value was obtained by Tukey-Kramer test for the interaction effect between time and group for diameter percent change in linear mixed model. 

Table 3. Vessel diameter changes between baseline and 12-week follow-up

 Nebivolol group Diltiazem group Combination group p-value

Changes in diameter, mm 1.44±0.28 2.04±0.60 1.30±0.47 <0.001

Percent changes in diameter, % 50.4±8.8a 67.8±12.8b 46.8±12.3a 0.0054

Values are presented as number of patients (%) or mean±standard deviation. CAG, coronary angiography; IC, intra-coronary; NTG, nitroglycerin; ACH, acetylcholine. a,b Tukey-Kramer test in linear mixed model were performed after adjusting for the effects of total cholesterol and LDL. Changes and percent changes in diameter is significantly greater in the Diltiazem group than the Nebivolol group or the Combination group.

 

Comment #9. 

Results, Coronary angiographic findings at baseline and at the 12-week follow-up, first paragraph, second sentence: “All patients had no significant coronary artery stenosis.” Please use a different word than “significant” as this is confusing in light its statistical meaning and usage.

Reply to Comment #9:

Thank you for your valuable comments and we agreed with your opinions. In order to avoid confusion, it has been modified with a clearer sentence. Kept these comments in mind, we revised the Results section in the manuscript accordingly. 

Page 16, Line 262-263

All patients had no fixed coronary artery lesion stenosis>50% in coronary angiography.

Comment #10. 

Results, Coronary angiographic findings at baseline and at the 12-week follow-up, first paragraph, third sentence: “12 weeks of random study medications…” what do the authors mean by “random study medications?”

Reply to Comment #10:

Thank you for your insightful comments we should address. 

The expression seems to have been ambiguous. It was intended to mean that the study drugs randomly assigned to each study group was taken for 12 weeks ((1) Nebivolol Group (5mg for 2 weeks/10mg for 10 weeks); (2) Diltiazem Group (90mg for 2 weeks/180mg for 10 weeks) ); (3) Low-dose Combination Group (2.5mg+45mg for 2 weeks/5mg+90mg for 10 weeks).The word “random” has been deleted to clarify the meaning. Kept these comments in mind, we revised the Result section in the manuscript accordingly. 

Page 16, Line 263-264

In the follow-up coronary angiography after 12 weeks of study medications, 

Comment #11. 

Results, Coronary angiographic findings at baseline and at the 12-week follow-up: The authors only provide the overall p-value for the global ANOVA test, which tells us whether there is or is not statistical evidence for a difference somewhere. They have omitted the important step of multiple comparisons, wherein they identify between which of the three groups a difference exists. Of course, this would mean that a significance level adjustment would need to be included, with a corresponding write-up in the Statistical Analysis section.

Reply to Comment #11:

Thank you for your critical comments and we totally agreed with your opinions. As the reviewer pointed out, re-analysis using a linear-mixed model was performed according to recommendations. Covariates included for fitting the model were study group, age, sex, primary and secondary endpoints, and baseline demographic variables that was less than p<0.1 when comparing between groups. 

As a result, changes in each group of coronary spasm corresponding to the primary endpoint and differences between groups were the same as in the previous analysis.

Kept these comments in mind, we revised the Result section and Figure 2 in the manuscript accordingly. 

Page 16, Line 257-262

In baseline coronary angiography, significant vasospasm after intracoronary acetylcholine administration in linear mixed model was observed in all three groups without statistical differences (percent changes in diameter in the Nebivolol Group vs. Diltiazem Group vs. Low-dose Combination Group: -87.9±4.9% vs. -86.4±6.3% vs. -85.5±5.7%, Nebivolol Group vs. Diltiazem Group p=0.977; Nebivolol Group vs. Low-dose Combination Group p=0.838; Diltiazem Group vs. Low-dose Combination Group p=0.997).

Page 16, Line 263-271

In the follow-up coronary angiography after 12 weeks of study medications, a significant improvement in coronary vasospasm was observed in all three groups (Nebivolol Group -37.5±6.4%, Diltiazem Group -18.8±9.4%, Low-dose Combination Group -38.9±8.6%, all p<0.001), but the magnitude of improvement in vasospasm was the greatest in the Diltiazem Group (percent changes in diameter from the baseline provocation test for Nebivolol Group vs. Diltiazem Group vs. Low-dose Combination Group: 50.4±8.8% vs. 67.8±12.8% vs. 46.8±12.3%, Nebivolol Group vs. Diltiazem Group p=0.008; Nebivolol Group vs. Low-dose Combination Group p=0.999; Diltiazem Group vs. Low-dose Combination Group p=0.017).

Figure 2

Comment #12. 

Table 3: Like Table 2, it is difficult to track change over time for any of these measures, which is the essential point of these secondary analyses.

Reply to Comment #12:

Thank you for your valuable comments that we should address. 

Re-analysis using a linear-mixed model was performed according to your recommendations. As a result, the Seattle Angina Questionnaire score corresponding to the secondary endpoint appeared to show significant improvement after study medication in overall study population, but baseline levels of overall Seattle Angina Questionnaire score between groups, change over time within groups, and difference regarding change over time between 3 groups were all insignificant. The table was modified according to the result of re-analysis and briefly changed, and the details were recorded in the S1 table.

Kept these comments in mind, we revised the Table 4 in the manuscript accordingly. 

 

Table 4. Changes in Seattle Angina Questionnaire during the 12-week follow-up

 Seattle Angina Questionnaire overall score 

 Baseline 12-week Changes p-valuea

Nebivolol 81.0±9.2 86.9±5.0 5.9±8.0 -

Diltiazem 80.6±7.9 82.5±10.4 2.2±9.3 -

Combination 79.5±7.2 86.6±6.9 7.1±8.1 -

Overall 80.3±7.9 85.4±7.8 5.2±8.5 0.0002

Values are presented as number of patients (%) or mean±standard deviation. a p-value was for the effect of time in linear mixed model after adjusting for the effects of sex and LDL. The effect of study group in linear model was not significant (p-value=0.824).

 

S1 Table. Details of changes in Seattle Angina Questionnaire during the 12-week follow-up

Baseline Nebivolol group (n=15) Diltiazem group (n=16) Combination group (n=17) p-value

Physical limitation

Score Sum 40.7±6.6 42.5±5.1 40.0±5.6 0.455

Percentile 47.4±27.3 47.7±23.9 39.7±27.0 0.611

Category 2.47±1.06 2.50±0.97 2.82±1.01 0.542

Angina frequency

Score Sum 8.5±2.0 8.8±2.0 9.1±1.7 0.633

Percentile 39.6±29.7 42.7±27.3 50.6±25.4 0.506

Category 2.80±1.21 2.63±1.15 2.24±1.15 0.378

Quality of life

Score Sum 11.4±4.7 10.4±3.2 10.9±3.2 0.774

Percentile 52.5±33.8 47.7±28.9 52.1±26.2 0.878

Category 2.87±1.41 3.13±0.96 2.77±1.15 0.670

Anginal stability

Score Sum 3.9±1.6 2.8±1.3 3.1±1.3 0.100

Percentile 54.7±31.3 35.1±25.0 38.1±25.5 0.110

Category 2.73±1.75 3.81±1.33 3.65±1.41 0.111

Treatment Satisfaction

Score Sum 16.5±2.4 16.1±1.6 16.4±2.6 0.829

Percentile 44.7±31.1 37.7±21.6 44.2±25.9 0.704

Category 2.67±1.18 2.88±0.96 2.59±0.94 0.713

Follow-up Nebivolol group (n=14) Diltiazem group (n=14) Combination group (n=17) p-value

Physical limitation

Score Sum 44.8±3.6 42.0±7.5 45.2±4.2 0.215

Percentile 49.1±25.6 36.2±22.4 52.3±25.2 0.178

Category 2.50±0.85 3.00±0.78 2.47±0.94 0.194

Angina frequency

Score Sum 10.7±0.7 9.8±1.6 10.0±1.5 0.167

Percentile 46.1±27.6 31.2±24.2 37.1±21.6 0.278

Category 2.86±1.10 3.43±0.85 3.06±0.90 0.283

Quality of life

Score Sum 10.8±1.7 10.5±1.9 10.3±2.3 0.793

Percentile 46.1±28.9 42.0±32.6 42.0±31.0 0.921

Category 2.64±1.08 2.86±1.23 2.77±1.15 0.886

Anginal stability

Score Sum 4.8±0.6 4.3±0.6 4.4±0.7 0.111

Percentile 42.8±25.2 24.3±21.3 31.2±22.3 0.110

Category 3.29±1.27 4.21±1.12 3.77±1.25 0.143

Treatment Satisfaction

Score Sum 15.9±1.5 15.9±2.4 16.7±2.0 0.419

Percentile 36.7±21.4 42.4±32.6 49.1±29.2 0.479

Category 2.93±1.00 2.64±1.34 2.53±1.18 0.637

Values are presented as number of patients (%) or mean±standard deviation.

 

Comment #13. 

Results, Changes in quality of life during 12-week follow-up, first paragraph, third sentence: We again have the phrase “12 weeks of random study medication.” Please clarify what “random study medication” means over 12 weeks.

Reply to Comment #13:

Thank you for your valuable comments and we sincerely sympathize with your opinions. 

The word “random” has been deleted to clarify the meaning. Kept these comments in mind, we revised the Result section in the manuscript accordingly.

Page 20, Line 292-294

After 12 weeks of study medications, a significant improvement in quality of life based on the overall Seattle Angina Questionnaire score from baseline was observed in the overall study population (p=0.0002).

Comment #14. 

Results, Changes in quality of life during 12-week follow-up: The authors either did not adjust the significance levels in the tests of change over time in each group for multiple comparison, or they did not report that they did so.

Reply to Comment #14:

Thank you for your valuable comments and we sincerely sympathize with your opinions. 

As the reviewer pointed out, re-analysis using a linear-mixed model was performed according to recommendations. As a result, the Seattle Angina Questionnaire score corresponding to the secondary endpoint appeared to show significant improvement after study medication in overall study population, but baseline levels of overall Seattle Angina Questionnaire score between groups, change over time within groups, and difference regarding change over time between 3 groups were all insignificant.

Kept these comments in mind, we revised the Results section and the Table 4 in the manuscript accordingly. 

Page 20, Line 291-296

Details regarding components of Seattle Angina Questionnaire are shown in S1 Table. After 12 weeks of study medications, a significant improvement in quality of life based on the overall Seattle Angina Questionnaire score from baseline was observed in the overall study population (p=0.0002). Baseline levels of overall Seattle Angina Questionnaire score between groups, change over time within groups, and difference regarding change over time between 3 groups were all insignificant.

Comment #15. 

Line 267 (blood pressure): “Random study medication” please explain what this means.

Reply to Comment #15:

Thank you for your valuable comments. 

As explained above, the expression “random” is removed to avoid ambiguity.

Kept these comments in mind, we revised the Result section in the manuscript accordingly. 

Page 21, Line 309-310

After 12 weeks of study medication, significant systolic blood pressure reduction from baseline was observed in the overall population (p<0.0001).

Comment #16. 

Lines 268-269 (blood pressure analyses): These p-values were not adjusted for multiple comparisons.

Reply to Comment #16:

Thank you for your insightful comments. 

The manuscript was modified according to the reanalysis result using the linear mixed model.

After 12 weeks of study medication, significant systolic blood pressure reduction from baseline was observed in the overall population (p<0.0001). Baseline levels of systolic pressure, diastolic pressure, and pulse pressure between groups, their changes over time within groups, and difference regarding changes over time between 3 groups were all insignificant.

Kept these comments in mind, we revised the Result section, Discussion section, Table 5 and S2 Table in the manuscript accordingly. 

Page 21, Line 307-312

Changes in systolic, diastolic and pulse blood pressures during the 12-week follow-up are shown at Table 5 and Fig 4. Details regarding blood pressures at 5 visits are shown in S2 Table. After 12 weeks of study medication, significant systolic blood pressure reduction from baseline was observed in the overall population (p<0.0001). Baseline levels of systolic pressure, diastolic pressure, and pulse pressure between groups, their changes over time within groups, and difference regarding changes over time between 3 groups were all insignificant.

Page 24, Line 342-343

(3) modest systolic blood pressure-lowering effects were observed in overall study population.

 

Table 5. Changes in blood pressures during the 12-week follow-up

 Nebivolol group Diltiazem group Combination group Overall population

Changes in SBP -8.5±24.6 -15.4±20.7 -12.5±15.9 -12.2±20.1

Changes in DBP -10.4±12.6 -5.2±10.5 -12.2±14.1 -9.5±12.7

Changes in pulse pressure 1.9±17.2 -10.1±16.6 -0.3±16.4 -2.7±17.1

Values are presented as number of patients (%) or mean±standard deviation. SBP, systolic blood pressure; DBP, diastolic blood pressure.

 

S2 Table. Details of changes in blood pressures during the 12-week follow-up

 Nebivolol group (n=15) Diltiazem group (n=16) Combination group (n=17) p-value

Visit 1 

SBP 132.1±18.5 136.2±15.3 137.9±15.7 0.602

DBP 76.3±9.1 81.9±8.9 83.7±12.5 0.124

Pulse pressure 55.9±18.2 54.3±16.1 54.2±11.3 0.946

Visit 2 

SBP 137.0±18.8 135.6±12.9 140.6±15.7 0.647

DBP 77.8±9.4 82.6±7.9 85.8±12.0 0.090

Pulse pressure 59.2±15.1 53.0±16.7 54.8±12.2 0.491

Visit 3 

SBP 128.4±16.4 129.3±19.4 129.9±12.0 0.964

DBP 77.7±10.4 80.0±8.7 83.3±11.3 0.305

Pulse pressure 50.7±10.2 49.3±13.5 49.6±11.9 0.621

Visit 4 

SBP 128.6±17.6 121.7±13.8 123.5±11.8 0.429

DBP 75.6±10.7 75.2±8.6 79.6±11.9 0.451

Pulse pressure 53.1±12.2 46.5±8.8 43.9±7.4 0.034

Visit 5 

SBP 124.2±20.5 121.9±15.4 125.4±17.8 0.859

DBP 66.1±12.4 76.2±8.5 71.5±10.9 0.054

Pulse pressure 58.1±18.0 45.6±11.8 53.9±18.1 0.131

Values are presented as number of patients (%) or mean±standard deviation. SBP, systolic blood pressure; DBP, diastolic blood pressure.

 

Comment #17. 

Lines 270-273: Same with DBP analyses, need to adjust p-values for multiple comparisons.

Reply to Comment #17:

Thank you for your valuable comments we should address. 

Please refer to the reply in Comment #16.

Comment #18. 

Discussion, Study Limitations: The authors state their study is explorative. Since they did not power their study for any particular hypothesis, I would agree and suggest that they bill this study as a pilot, including in the title.

Reply to Comment #18:

Thank you for your valuable comments and we agreed with your opinions. In our study, the sample calculation was impossible because this study was a pilot study dealing with a topic that had not been studied previously. Therefore, patients were recruited for a fixed period of time (January 2018 to March 2019) and registration was then terminated. To clarify that this study is a pilot study, the title of this study has been modified. We additionally stated in the limitation section that sample calculation was impossible because this study was a pilot study.

Kept these comments in mind, we revised the Title, Introduction, Methods, and the Discussion section in the manuscript accordingly. 

Title

Comparison of Nebivolol versus Diltiazem in improving coronary artery spasm and quality of life in patients with hypertension and vasospastic angina: A prospective, randomized, double-blind pilot study

Page 6, Line 81-84

Therefore, we performed a prospective, randomized, double-blind pilot study to compare nebivolol versus diltiazem in improving coronary artery spasm and quality of life in hypertensive patients with vasospastic angina during a 12-week follow-up.

Page 13, Line 231-233

To the best of our knowledge, this was the first pilot study to compare nebivolol versus diltiazem in improving coronary artery spasm; therefore, the sample size could not be calculated from previously published data.

Page 24, Line 335-337

In this prospective, randomized, double-blind pilot study, we compared the effects of nebivolol and diltiazem on improvements in coronary artery vasospasm, quality of life, and blood pressure in hypertensive patients with documented coronary vasospasm.

Page 26, Line 393-396

This study was a pilot study that confirmed the vasodilating effect of nebivolol in an in-vivo setting; therefore, sample calculation was unavailable for the study. The number of participants and follow-up duration was limited. A clinical trial with a larger number of participants based on adequate sample calculation and long-term follow-up is warranted.

---

## [Decision Letter · Decision Letter 1]

31 Aug 2020

Comparison of Nebivolol versus Diltiazem in improving coronary artery spasm and quality of life in patients with hypertension and vasospastic angina: A prospective, randomized, double-blind pilot study

PONE-D-20-15573R1

Dear Dr. Hong,

We’re pleased to inform you that your manuscript has been judged scientifically suitable for publication and will be formally accepted for publication once it meets all outstanding technical requirements.

Kind regards,

Yoshihiro Fukumoto

Academic Editor

PLOS ONE

Additional Editor Comments (optional):

Reviewers' comments:

Reviewer's Responses to Questions

**Comments to the Author**

1. If the authors have adequately addressed your comments raised in a previous round of review and you feel that this manuscript is now acceptable for publication, you may indicate that here to bypass the “Comments to the Author” section, enter your conflict of interest statement in the “Confidential to Editor” section, and submit your "Accept" recommendation.

Reviewer #1: (No Response)

Reviewer #2: All comments have been addressed

Reviewer #3: All comments have been addressed

Reviewer #4: All comments have been addressed

2. Is the manuscript technically sound, and do the data support the conclusions?

Reviewer #1: Yes

Reviewer #2: Yes

Reviewer #3: Yes

Reviewer #4: (No Response)

3. Has the statistical analysis been performed appropriately and rigorously? 

Reviewer #1: Yes

Reviewer #2: Yes

Reviewer #3: Yes

Reviewer #4: (No Response)

4. Have the authors made all data underlying the findings in their manuscript fully available?

Reviewer #1: Yes

Reviewer #2: Yes

Reviewer #3: Yes

Reviewer #4: (No Response)

5. Is the manuscript presented in an intelligible fashion and written in standard English?

Reviewer #1: Yes

Reviewer #2: Yes

Reviewer #3: Yes

Reviewer #4: (No Response)

6. Review Comments to the Author

Reviewer #1: The authors responded to the reviewer’s remarks in detail. The manuscript was improved almost satisfactorily.

Reviewer #2: (No Response)

Reviewer #3: (No Response)

Reviewer #4: (No Response)

7. PLOS authors have the option to publish the peer review history of their article (what does this mean?). If published, this will include your full peer review and any attached files.

Reviewer #1: No

Reviewer #2: No

Reviewer #3: No

Reviewer #4: No

---

## [Editor Report · Acceptance letter]

2 Sep 2020

PONE-D-20-15573R1 

Comparison of Nebivolol versus Diltiazem in improving coronary artery spasm and quality of life in patients with hypertension and vasospastic angina: A prospective, randomized, double-blind pilot study 

Dear Dr. Hong:

I'm pleased to inform you that your manuscript has been deemed suitable for publication in PLOS ONE. Congratulations! Your manuscript is now with our production department. 

Kind regards, 

on behalf of

Dr. Yoshihiro Fukumoto 

Academic Editor

PLOS ONE